# Reliability of high-quantity human brain organoids for modeling microcephaly, glioma invasion and drug screening

Anand Ramani[1], Giovanni Pasquini[2], Niklas J. Gerkau[3], Vaibhav Jadhav[1], Omkar Suhas Vinchure[1], Nazlican Altinisik[1], Hannes Windoffer[1], Sarah Muller[1], Ina Rothenaigner[4], Sean Lin[4], Aruljothi Mariappan[1], Dhanasekaran Rathinam[1], Ali Mirsaidi[5], Olivier Goureau [6], Lucia Ricci-Vitiani[7], Quintino Giorgio D'Alessandris[8], Bernd Wollnik [9], Alysson Muotri[10,11], Limor Freifeld[12], Nathalie Jurisch-Yaksi [13], Roberto Pallini[8], Christine R. Rose[3], Volker Busskamp [2], Elke Gabriel[14], Kamyar Hadian [4,15] & Jay Gopalakrishnan[1,15] ✉

Brain organoids offer unprecedented insights into brain development and disease modeling and hold promise for drug screening. Significant hindrances, however, are morphological and cellular heterogeneity, inter-organoid size differences, cellular stress, and poor reproducibility. Here, we describe a method that reproducibly generates thousands of organoids across multiple hiPSC lines. These High Quantity brain organoids (Hi-Q brain organoids) exhibit reproducible cytoarchitecture, cell diversity, and functionality, are free from ectopically active cellular stress pathways, and allow cryopreservation and re-culturing. Patient-derived Hi-Q brain organoids recapitulate distinct forms of developmental defects: primary microcephaly due to a mutation in CDK5RAP2 and progeria-associated defects of Cockayne syndrome. Hi-Q brain organoids displayed a reproducible invasion pattern for a given patient-derived glioma cell line. This enabled a medium-throughput drug screen to identify Selumetinib and Fulvestrant, as inhibitors of glioma invasion in vivo. Thus, the Hi-Q approach can easily be adapted to reliably harness brain organoids' application for personalized neurogenetic disease modeling and drug discovery.

Advancements in three-dimensional (3D) culturing of pluripotent stem cells and tissue engineering have led to the generation of 3D brain organoids, which offer unique opportunities to recapitulate various aspects of brain development and disease[1–4]. In particular, brain organoids derived from healthy hiPSC have modeled early embryonic brain development[5–7]. Brain organoids generate more advanced cell types and functionally active neuronal networks when cultured long-term[3,8]. Besides recapitulating development, brain organoids hold value in modeling genetic brain disorders when using patient-specific hiPSC or perturbing target genes by genome editing

technologies[2,9,10]. If generated on a large scale with reproducible quality, brain organoids can also offer drug screening assays in 3D tissues to identify therapeutic compounds directly on disease-relevant human tissues.

Several methods have been developed to culture brain organoids tailored to specific questions of interest. However, regardless of the protocol, significant challenges limit their application for reliable disease modeling and drug screening. These include morphological and regional heterogeneity, inter-organoid size differences, ectopically induced activation of cellular stress pathways, and, most importantly,

the limited quantity of organoids per batch, which limits the generation of statistically significant data[11,12]. To date, there is no standard method for generating reproducible qualities of hiPSC-derived human brain organoids with regulated cell diversity, devoid of excessive cellular stress pathways to reliably model disorders and simultaneously allow upscaling for drug screening approaches.

Most methods of brain organoid differentiation include the embryoid body (EB). EBs are 3D pluripotent cell aggregates that mimic some intermediate structure of the developing embryo that can differentiate into cells of all three germ layers[13]. Typically, EB is generated manually or using microwell plates from disassociated pluripotent cells with or without embedding in an extracellular matrix[14,15]. The neuroectoderm is then differentiated from EB[16,17]. The differentiation outcome depends on the quality of EBs, which is determined by cell number, medium conditions, EB sizes, and viability. For example, differing EB sizes can impact the cell diversity in their terminal differentiation[18–20]. These variables, all together, can cause size differences between organoids, morphology, and cellular diversity and can significantly limit the number of organoids one can generate per batch.

Here, we describe a simplified culturing method to induce direct differentiation of hiPSC into the neural epithelium, omitting the EB stage and the use of an extracellular matrix. Notably, by using custom-designed, coating-free, pre-patterned microwells, we have complete control over the sizes of early-stage neurospheres. Following a quick transfer to spinner-flask bioreactors, our method generates brain organoids in high quantities (referred to hereafter as Hi-Q brain organoids). This Hi-Q approach is highly efficient as we can culture several hundreds of brain organoids within a batch from multiple hiPSC lines. Furthermore, Hi-Q brain organoids exhibit similar cell diversity, cytoarchitectures, maturation time, and functionality. Importantly, brain organoids generated with this platform displayed minimum to no ectopically activated cellular stress pathways, which has been previously shown to impair cell-type specification[21]. We report that Hi-Q brain organoids can also be successfully cryopreserved and re-cultured.

Human brain organoids offer the unique opportunity to underpin the pathomechanisms of neurogenetic diseases, which are challenging to model in 2D or using non-human models such as rodents[9,22,23]. However, no standard method is optimized to robustly model distinctly differing genetic brain diseases. Hi-Q brain organoids could recapitulate the phenotypes of both primary microcephaly due to a mutation in the centrosomal protein CDK5RAP2 and the neurological phenotype of progeria-associated Cockayne syndrome due to DNA damage. Finally, to assess the applicability of Hi-Q brain organoids for drug screening in 3D tissues, first, we modeled glioma invasion by fusing patient-derived glioma stem cells (GSCs) to Hi-Q brain organoids. We ultimately used the GSC-invading Hi-Q brain organoids for a medium-throughput drug screening. Applying machine-learned algorithms and automated imaging on our Hi-Q organoids, we identified Selumetinib and Fulvestrant as potent GSC invasion inhibitors in both in vitro and mouse in vivo glioma xenografts.

## Results

### Generation of Hi-Q brain organoids

Conventionally, most unguided brain organoid differentiation methods include embedding disassociated hiPSC with or without an extracellular matrix and processing them into the EBs before differentiation into the neurospheres[5,6,24,25]. We previously demonstrated a method of directly inducing differentiation into neural epithelium from hiPSC. These brain organoids expressed various ranges of cell types, electrically active neural networks, optic vesicles, and cell types derived from surface ectoderm[3,26,27]. This protocol involved the manual processing of neurospheres. As objects in culture dishes, neurospheres tend to exhibit variable shapes and sizes, and most importantly, it limits the number of organoids one can generate per batch.

We reasoned that differentiating hiPSC directly into neurospheres in a confined space could restrict heterogeneity and generate uniform-sized brain organoids in large quantities with increased homogeneity and similar cell diversity. To do this, we exposed disassociated hiPSC directly to a neural induction medium (Supplementary Table 1) to induce the neurospheres in microwells equipped with a round bottom satisfying two prerequisites. First, microwells allow identical diffusion conditions for all spheres and unique physiological sphere formation. Second, microwell material does not require precoating, which could dampen the cells' attachment or a centrifugation step, forcing cell pelleting. For this purpose, we fabricated a custom-designed spherical plate using a medical-grade, inert Cyclo-Olefin-Copolymer (COC), which offers an ideal surface property. The plate consists of 24 large wells, which are micropatterned to contain 185 equally sized microwells of 1x1mm at the opening and 180 μm in diameter at their round base (Fig. 1A). Typically, each microwell exhibits an inverted pyramid shape with a rounded bottom. This geometry induces the seeded cells to form spheres through mutual adhesion.

We noticed that hiPSC readily settled within a day of plating in our spherical plate, even without a centrifugation step. This suggests that our spherical plate may provide a more suitable environment for sphere formation (Fig. 1B). Typically, using our spherical plates, we could differentiate 10,000 hiPSC into uniform-sized 3D neurospheres within each microwell. While using a Rho-kinase (ROCK) inhibitor at this stage will alleviate cell death, prolonged exposure could change the cell's metabolism and induce the meso-endodermal differentiation pathway[28,29]. Indeed, prolonged use of ROCK inhibitors is associated with generating organoids with ectopically active cellular stress pathways[21]. Therefore, after 24 h of initial culturing in a neural induction medium, we omitted the ROCK inhibitor.

On day 5, we noticed uniform-sized neurospheres. Imaging them revealed that they are highly similar from well to well, exhibiting characteristic neural rosette organization with primary cilia emanating apically into the lumen. Furthermore, neurosphere size remained consistent across several independent batches (Fig. 1C, E). We then transferred these uniform-sized Matrigel-free neurospheres to spinner bioreactors containing 75 ml neurosphere medium (Fig. 1D, E, F, Supplementary Table 1). After culturing for four days, we switched to a brain organoid differentiation medium containing 5 μM SB431542 and 0.5 μM Dorsomorphin, inhibitors of TGF-β and BMP pathways to initiate an undirected neural differentiation. Twenty-one days later, we switched to a brain organoid maturation medium and cultured organoids until day 150 with a constant spinning rate of 25 RPM without noticing disintegrated spheroids (Supplementary Movie 1).

To assess the overall versatility of the Hi-Q approach, we generated organoids from six independent hiPSC lines (Four healthy and two derived from microcephaly patients). Notably, in this approach, the organoids grew in size progressively over time (Fig. 1G) (except organoids derived from microcephaly patients, which are described later). In this Hi-Q platform, we generated ~15,373 organoids across 39 batches (Supplementary Table 2). Measuring 300 randomly selected Hi-Q brain organoids across four hiPSC lines revealed that organoid size was highly consistent within a batch and across hiPSC lines (Fig. 1H). Furthermore, the organoids showed a consistent and proportional size increase from day 20 to 60 across all hiPSC lines. This indicates that organoids do not aberrantly vary in growth with our Hi-Q approach (Fig. 1I). In all cases, the organoids displayed high integrity as we detected only one or two disintegrated organoids in a batch of 300 (Fig. 1J). These results suggest that the Hi-Q approach is robust, versatile, and easy to handle.

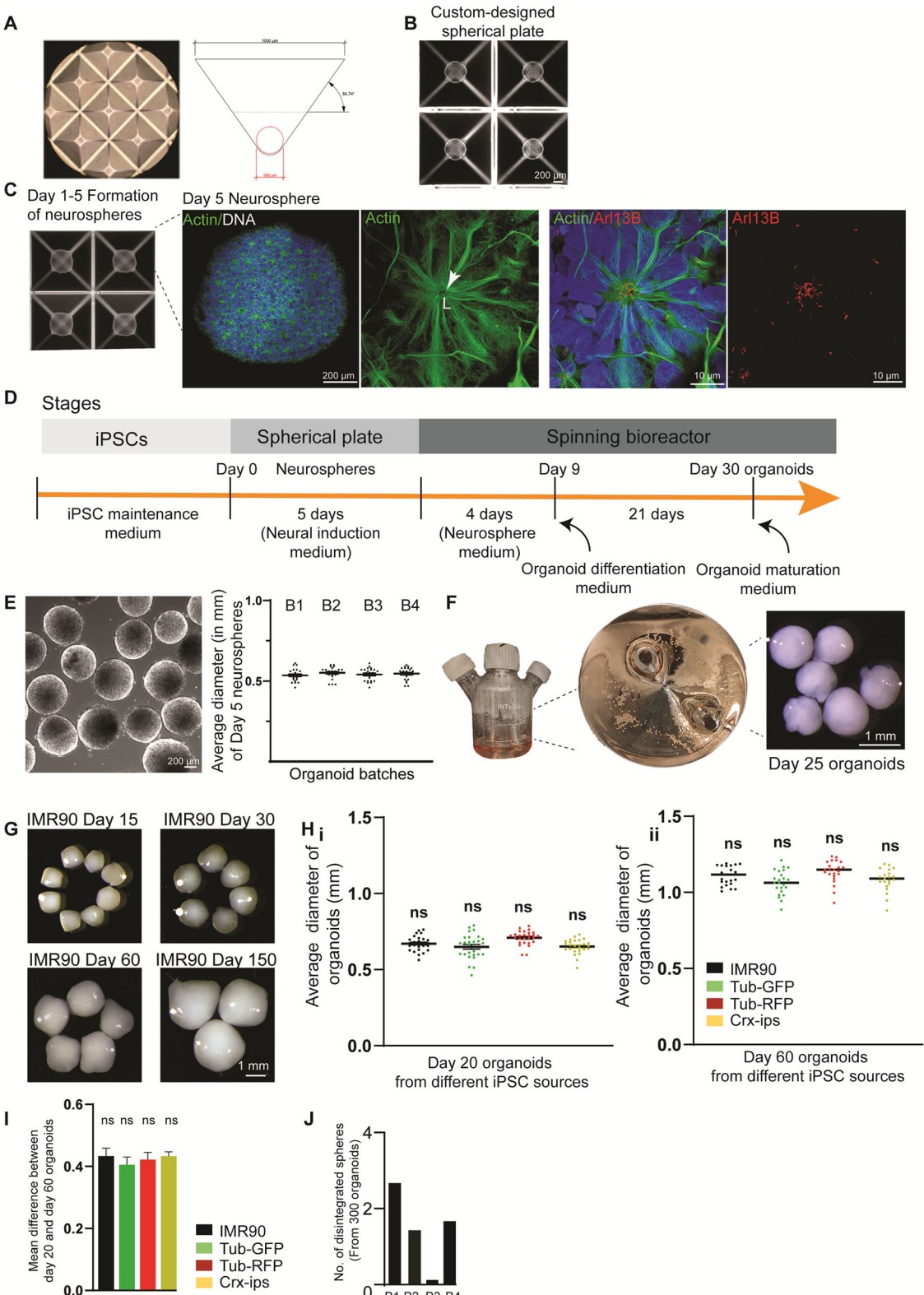

## Time-resolved single-cell RNA-sequencing of Hi-Q brain organoids reveals similar cell diversities and is free from ectopic stress-inducing pathways

To dissect the cell diversity of Hi-Q brain organoids and correlate it to the human brain, we performed single-cell RNA-sequencing (scRNA-seq). First, to test similarities across independent Hi-Q brain organoids, we sequenced and compared Day 25 Hi-Q brain organoids across three independent batches (Supplementary Fig. 1). To assess the similarities between the batches, we standardized the comparison using the 2000 most highly variable genes across the batches. We then applied *the k*-nearest neighbor network (*k*nn) approach to analyze distances and progression of transcriptional changes among cells in the 2D representation[30,31]. The *k*nn approach used a PCA embedding (Supplementary Fig. 1A, B). In this analysis, the cells did not cluster by batches, suggesting the degree of similarity is high, or the presence of low covariance across the batches,

**Fig. 1 | Generation of Hi-Q brain organoids. A** Microscopic view showing a custom-designed spherical plate with microwells and schematic view of the inverted pyramid-like microwell. The angles and measurements are shown. **B** hiPSC settles and readily forms spheres in the spherical plate's microwells. The panel shows the scale bar. **C** Formation of neurospheres in the microwells of the spherical plate. The magnified image at the right shows a representative neurosphere. Magnified panels show a neural rosette stained with actin (green) with primary cilia emanating into the lumen (L) at the apical side of the rosette marked by Arl13B (Red). Panels show the scale bar. **D** Different stages of Hi-Q brain organoid generation. **E** A group of neurospheres. The graph at right shows no significant difference between the organoid diameter across four independent batches. At least twenty (*n* = 20) randomly chosen neurospheres were measured from each batch. Statistical analysis was done using one-way ANOVA, followed by Tukey's multiple comparisons test. Data presented as mean ± SEM. The cell line used is IMR90. The panel shows the scale bar. **F** Maturation of neurospheres into Hi-Q brain organoids in spinner flasks. Macroscopic images show a group of organoids. The panel shows

the scale bar. **G** Hi-Q brain organoids increase in size progressively over time from day 15 to day 150. The panel shows the scale bar. **H** The average diameter of twenty-day (**i**) and fifty-day (**ii**) old organoids were differentiated from four independent hiPSC lines. Note that there is no significant difference among the different hiPSC donors within each time point. At least twenty-five (*n* = 25) randomly chosen organoids were analyzed across three independent batches (*N* = 3). Statistical analysis was done using one-way ANOVA, followed by Tukey's multiple comparisons test. Data presented as mean ± SEM. **I** The graph shows no size difference between twenty-day and sixty-day-old organoids, showing a regulated growth rate between two-time points. At least twenty-five (*n* = 25) randomly chosen organoids were analyzed across three independent batches (*N* = 3). Data distribution is given in the panel (**H**). Statistical analysis was done using one-way ANOVA, followed by Tukey's multiple comparisons test. Data presented as mean ± SEM. **J** The bar diagram quantifies the number of disintegrated organoids in each batch. At least three hundred (*n* = 300) organoids were randomly sampled across four independent batches (*N* = 4).

and hence no correction was required for batch-to-batch variation. To ease the analysis of the similarities between batches, we chose three major cell types: progenitors (based on SOX2, GLI3, and PAX6), cycling progenitors (based on MKI67, CENPF, and NUSAP1), and early neurons (based on DCX, NCAM, and GAP43). We then analyzed the differences between the proportions of cells in each cell type across the batches (Supplementary Fig. 1C–E). This analysis did not yield significant differences, indicating a low batch-to-batch variation in cell types and proportions in our Hi-Q brain organoids.

To dissect the cell diversity changes across various age groups, we analyzed 16,228 cells isolated from three organoids at each time of day: 60, 90, and 150. The sequenced cells belonged to cell types such as proliferating radial glia (Pro-RG) cells expressing TTYH1, intermediate precursor cells (IPC) expressing MKI67 and NUSAP1, inhibitory neurons (IN) expressing GAD2, and excitatory neurons (EN) expressing NEUROD2 and NEUROD6 (Fig. 2A, Supplementary Fig. 2A–C). We then applied *the k*nn approach to analyze distances and progression of transcriptional changes. All three stages of organoids comprise structural, immature, proliferating, and neuronal populations (Fig. 2A–C). Notably, cells annotated as proliferating radial glia cells (RG) and IPCs correspond to the cells in the S and G2/M phases (Supplementary Fig. 2D, E).

We then used the annotated cell types to evaluate the degree of similarities and maturation at each stage of Hi-Q brain organoids. First, we noticed that individual organoids showed similar cell proportions at the same age. Next, as the organoids progressively matured, we observed an accumulation of mature neuronal types, especially excitatory neurons (EN), in day 90 and 150 organoids (Fig. 2B). We then applied principal component analysis to assess the organoids' similarities at the transcriptional level within an age group. Importantly, this analysis revealed that the cellular composition of individual organoids exhibited a high similarity with organoids from the same age groups and widely differed from other age groups, indicating that the Hi-Q approach generates highly reproducible organoids and is reliable (Supplementary Fig. 2F). Analyzing the cell types computed from the -*k*nn analysis that integrated cell diversities of all three age groups of organoids highlighted the presence of apical neural progenitors (AP), radial glia (RG), proliferating radial glia (Pro-RG), astrocytes and oligodendrocytes (A/O), intermediate precursor cells (IPC), developing neurons (Dev-N), developing hindbrain (Dev-HB), early neurons (EaN), excitatory neurons (EN) and inhibitory neurons (IN) (Fig. 2C, D).

To further analyze the developmental trajectories of terminally developed populations within the dataset, we used a pseudotime analysis based on diffusion from the RG cluster (Fig. 2E). The gene expression patterns across the two trajectories agree with the unbiased, calculated markers from the cell types. We observed mainly

two trajectories. The first trajectory (blue line) expressing markers of dorsal telencephalon development (GLI3, EOMES, and GRIA2) leads to the formation of ENs. The second trajectory (green line), expressing markers of ventral telencephalon development (CCND2, DLX5, GAD2)[30], concludes with the appearance of INs (Fig. 2E and Supplementary Fig. 2G, H).

Recent work reported that activation of cellular stress pathways in organoids interferes with the developmental process required to generate distinct cell identities of the human brain[21]. Thus, optimizing a method that allows the generation of brain organoids free from cellular stress pathways is critical. To test if Hi-Q brain organoids are free from those ectopic stress-inducing pathways, we analyzed the expression level of previously described stress markers PGK1, ARCN1, and GORASP2[21]. Notably, Hi-Q brain organoids showed a lower stress marker expression than published brain organoid datasets but slightly higher than adult human brain datasets[21,30] (Fig. 2F). This analysis indicates that our in vitro culture does not yet meet the natural conditions mimicking brain development in vivo. Yet, it is relatively free from ectopically activated cellular stress pathways, which impairs cell-type specification. In summary, our temporally resolved sc-RNA-sequencing data reveal that the Hi-Q approach can generate brain organoids with reduced levels of culture-induced stress and reproducible levels of cell diversity closer to an in vivo situation.

We then analyzed whether Hi-Q brain organoids exhibit cell types comparable to those generated through EB formation. To this end, we computed the major cell types and their proportions of age-matched Hi-Q brain organoids and EB-based brain organoids. To compare 60-day-old organoids, we used raw sc-RNA counts and gene expression matrices from a published study[32]. For comparing 90- and 150-day-old organoids, we used a data set from a different study that offered processed sc-RNA data from 90- and 180-day-old organoids[7]. Notably, these studies emerged from donor hiPSC that differ from ours. Our integrated analysis identified six major cell types: neurons (DCX, or STMN2), radial glia (SOX2, PAX6, HES1, GLI3, or TTYH1), proliferating radial glia (MKI67, TOP2A, NUSAP1, or CENPF), astrocytes (S100B, GFAP, AQP4, ID3, IL33, or GJA1), and inhibitory (GAD2, GAD1, SIX3, DLX6-AS1, DLX1, DLX2, DLX5, SCGN, or ISL1) and excitatory neurons (SLC17A7, SATB2, NEUROD2, or EMX1), cell lineages emerge from both ventral and dorsal brain in both organoid groups (Supplementary Fig. 3A). When calculating the relative proportions of the cell types, our analysis revealed that organoids differentiated via EBs have abundant astrocytes. On the other hand, Hi-Q brain organoids contained an increased proportion of proliferating cell types (Radial glia and proliferating radial glia) (Supplementary Fig. 3B). These findings suggest that although Hi-Q brain organoids harbor similar cell types of EB-based brain

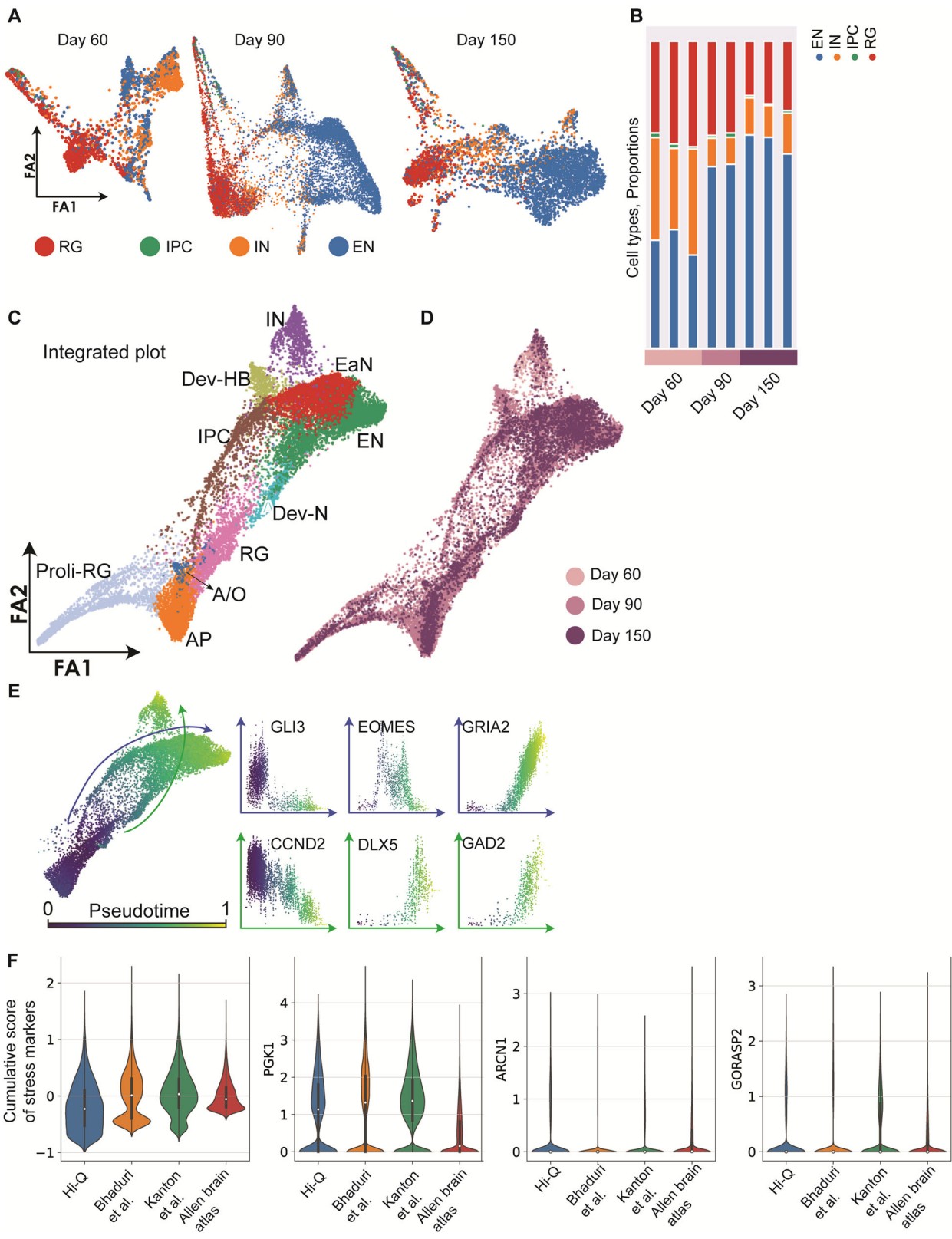

organoids, they contain more proliferative cell populations and are slightly behind the maturation status of those emerging from EBs.

**Hi-Q brain organoids progressively develop over time and reveal mature neuronal markers.** Next, we analyzed the cytoarchitecture of the organoids using histology. We selected a panel of markers for each cell type in organoids derived from the IMR90 on day 20 and day 60 and quantified them. Our selected panel included Nestin and SOX2 (progenitors), DCX (early neurons), Acetylated α-tubulin (neurons and cilia), Arl13b (cilia), Actin (neurons), TUJ-1 (pan-neuronal), MAP2 (cortical neurons), CTIP2 (layer 4 and 5-specific), Tau (cortical neurons), PCP4 (Purkinje neurons), GAD67 (glutamatergic neurons),

**Fig. 2 | Cell type diversity in Hi-Q brain organoids across maturation. A** Force Altas (FA) 2D representation of the neighbor graph of Hi-Q brain organoids at different time points of development (Day 60, 90, and 150). Cells are colored according to the label with the highest score given by the Lasso logistic regression model trained on primary brain data[31]. **B** Histogram plot reporting the proportion of different cell types at different time points of development. **C, D** FA representation of the neighbor graph of the integrated datasets from all stages. In panel (**C**), clusters are annotated using the expressed gene markers: Pro-RG, proliferating radial glia; AP, apical progenitors; RG, radial glia; A/O, astrocytes, and oligodendrocytes; IPC, intermediate precursor cells; Dev-N, developing neurons; Dev-HB, developing hindbrain; EaN, early neurons; EN, excitatory neurons; IN, inhibitory neurons. In panel (**D**), cells are colored according to the time they were sampled. **E** The subset of the whole integrated dataset, except the pro-RG cluster, colored by pseudotime, showing the two trajectories of developing IN (green arrow) and EN (blue arrow). The scatter plots show the expression range of characteristic IN and EN genes for each cell in function of the pseudotime. **F** Violin plots compare the level of expression of PGK1, ARCN1, and GORASP2, as well as the cumulative score between Hi-Q brain organoids (nine organoids from three different age groups used in Fig. 2B), other brain organoids[21,30], and primary data from the literature. The overlaid box indicates median, quartiles, and 1.5× IQR whiskers; outliers are individual points.

Synapsin-1 (presynaptic), and PSD95 (post-synaptic). As previously described, we conducted whole-mount immunostaining and confocal imaging of intact organoids after tissue clearing to preserve organoid integrity[33,34].

Day 20 organoids mainly exhibited early developmental markers of progenitors and early neurons and barely showed distinct cortical plates decorated by mature neuronal markers. Neuronal markers such as MAP2 and TUJ-1 were expressed but only localized in the cell body and did not entirely segregate into axons, indicating that these organoids are at the early stage of neuronal differentiation (Fig. 3A). In contrast, day 60 organoids were more prominent in size and exhibited an enriched level of mature neuronal markers with distinguishable cortical plates of similar thickness. Likewise, the proportions of cortical and layer-specific neurons increased with time, with Tau, PCP4, PSD95, Synapsin 1, and CTIP2-positive cells increasing in day 60 organoids (Fig. 3B–E, Supplementary Fig. 4B, and Supplementary Movie 2–14). Finally, to test if Hi-Q brain organoids could show different layers of cortex organization, such as ventricular zone/subventricular zone (VZ/SVZ) and outer SVZ (oSVZ), we tested and quantified for the expression of respective identity markers in our scRNA seq data. These included VZ markers (such as EMX2 and PAX6), SVZ markers (TBR2), and oSVZ markers (GFAP, TNC, PTPRZ1, FAM107A, HOPX, and LIFR)[35] (Supplementary Fig. 5A–C). To experimentally validate the presence of these cell types, we stained the organoid slices for SVZ (TBR2) and oSVZ (PTPRZ1 and phospho-Vimentin) markers (Supplementary Fig. 5D–F). These data indicate that whole-brain organoids cultured via the Hi-Q approach generate cell types of the early brain and differentiate into mature cell types distinct from the early brain's germinal zones.

## Spontaneous activity and neuronal networks in Hi-Q brain organoids

Next, we investigated whether Hi-Q brain organoids exhibit functional activity and form active neuronal networks. A reliable indicator for neuronal activity and network formation in the developing and mature brain is intracellular $Ca^{2+}$ signaling[36]. To probe for functional activity, we performed imaging of intracellular $Ca^{2+}$ in brain organoids at days 30, 40, 50, and 150. To this end, we loaded the organoids with the calcium indicator dye Oregon Green BAPTA1-AM (OGB-1) by bolus injection, staining essentially all cell bodies in the field of view (Fig. 4A). Live imaging of OGB-1 fluorescence revealed vivid spontaneous intracellular $Ca^{2+}$ signaling in all preparations and stages analyzed (Fig. 4B), with roughly half of all cells (42–54%) being active in the investigated age groups (Fig. 4C). $Ca^{2+}$ signals were variable in amplitude and duration. We detected both single transients, lasting 10–20 s, and burst-like events, lasting 1–2 min, at each stage. Moreover, ultra-slow $Ca^{2+}$ fluctuations were observed, to which faster events usually added on (Fig. 4B). The activity was not generally synchronized across the cells in the field of view, although single events occasionally occurred in several cells simultaneously (see asterisks in Fig. 4B).

In organoids on days 40, 50, and 150, spontaneous activity was strongly dampened ($p = 1.39E\text{-}14$; $p = 7.96E\text{-}34$; $p = 3.88E\text{-}48$) in the presence of tetrodotoxin (TTX; $1\,\mu M$) that blocks voltage-gated $Na^+$ channels, suggesting that $Ca^{2+}$ signals were mostly secondary to action potential generation and activation of voltage-dependent $Ca^{2+}$ channels (Fig. 4D). Next, we applied the neurotransmitters glutamate and GABA by bath perfusion for 10 s to further characterize functional activity in brain organoids. Notably, in day 40, day 50, and 150-day-old organoids, all cells tested responded to the application of 1 mM glutamate (day 40: $N = 3$, $n = 200$; day 50: $N = 3$, n = 142; day 150: $N = 3$, $n = 230$) or 1 mM GABA (day 40: $N = 3$, n = 173; day 50: $N = 3$, n = 211; day 150: $N = 3$, n = 218) with large and long-lasting $Ca^{2+}$ transients (Fig. 4E, F). Glutamate-induced $Ca^{2+}$ signals at day 50 were significantly reduced by the application of blockers of the NMDA-receptor blocker APV ($100\,\mu M$) ($N = 3$, $n = 133$; $p = 2.84E\text{-}12$), and further dampened by combined application of APV with the AMPA-receptor blocker NBQX ($50\,\mu M$) ($N = 3$, $n = 108$; $p = 3.50E\text{-}29$) (Fig. 4E, F). $Ca^{2+}$ transients induced by GABA application were significantly reduced upon inhibition of action potential generation by TTX ($N = 3$, $n = 204$; $p = 3.85E\text{-}14$), and nearly completely suppressed by additional perfusion with NiCl2. Taken together, these results indicate that Hi-Q brain organoids develop functionally active neural networks with cells expressing voltage-gated, TTX-sensitive $Na^+$ channels and voltage-gated $Ca^{2+}$ channels, major hallmarks of differentiated neurons. Moreover, cells respond to glutamate and GABA, the brain's primary neurotransmitters. Using pharmacological tools, we also obtained evidence for the functional expression of ionotropic glutamate receptors, namely AMPA- and NMDA receptors. Finally, the sensitivity of GABA-induced $Ca^{2+}$ signals to TTX, and $NiCl_2$ indicates that $Ca^{2+}$ transients evoked by bath application GABA were secondary to cellular depolarization upon activation of ionotropic $GABA_A$ receptors[37,38].

## Hi-Q brain organoids can be cryopreserved, thawed, and re-cultured

Unlike patient-derived liver and intestine organoids, brain organoids have not been cryopreserved, thawed, and re-cultured, an aspect that limits the flexible and economical use of brain organoids[39,40]. We could successfully freeze 18-day-old Hi-Q brain organoids and cryopreserve them in liquid nitrogen. After three months of cryopreservation, we re-cultured them after thawing and analyzed them for recovery (Fig. 5A, see "Method" section). Notably, 18-day-old organoids predominantly contain undifferentiated proliferating SOX2-positive progenitors exhibiting TUJ-1-positive primitive cortical plate. To re-culture the organoids, we first thawed them into the organoid culturing medium for 48 h in a stationary petri dish and then cultured them in spinner flasks for several days (Fig. 5B). Although 48 h post-thawing, organoids did appear slightly deformed, twelve days after re-culturing (30-day-old organoids), the organoids were morphologically similar to control organoids that had never been frozen. Notably, thawing and re-culturing did not cause a significant disassociation or change in size (Fig. 5C, D). Importantly, we could also thaw and culture organoids in a petri dish without involving a spinner flask. Although the initial growth rate of thawed organoids was slower than that of the controls, they followed a growth profile similar to that of the control organoids

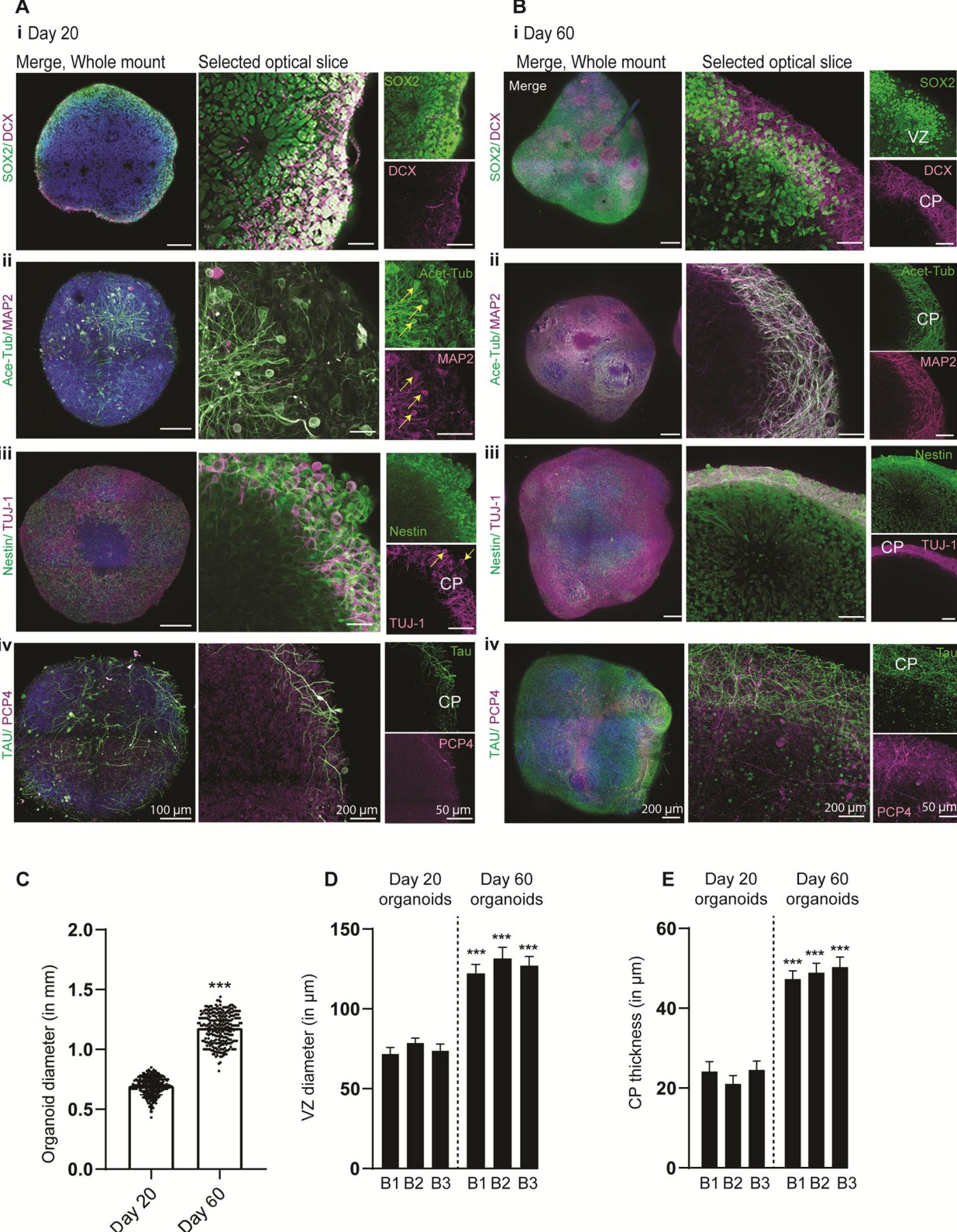

(Supplementary Fig. 6A, B). Calculating the percentage recovery, we could obtain at least 75% of the recovery after thawing. In most cases, we could recover at least 90% of the organoids (Supplementary Fig. 6C). The differences in the frequencies of dead cells and cytoarchitecture between 48 hrs and 10 days after the thawing of organoids indicated the successful recovery of surviving cells after cryopreservation (Supplementary Fig. 6D, E).

We then analyzed the integrity and composition of cytoarchitectures after thawing and re-culturing. We immunostained them for SOX2, TUJ-1, and TUNEL, which labels NPCs, primitive cortical plates, and dead cells, respectively. Our analysis revealed that the re-cultured organoids exhibit strikingly similar cytoarchitectures to age-matched control groups, which were not cryopreserved and re-cultured. Notably, there was no difference in the frequencies of TUNEL-

**Fig. 3 | Hi-Q brain organoids mature over time. A, B** Tissue clearing and whole-mount staining of day 20 (**Ai-iv**) and 60-day (**Bi-iv**) old organoids show that Hi-Q organoids mature over time. SOX2 and Nestin label NPCs at the ventricular zone (VZ), DCX, Acetylated tubulin, MAP2, TUJ-1, PCP4, and Tau mark neurons at the cortical plate (CP). Note cell body localization of acetylated α-tubulin, MAP2, and TUJ-1 (Arrows in **Aii-iii**) on day 20, organoids are remodeled into defined cortical plates (CP) on day 60 organoids (**Bi-iii**). Likewise, the cortical neuronal markers Tau and PCP4 were remodeled into distinct cortical plates in day 60 Hi-Q brain organoids (**Aiv-Biv**). Therefore, CPs are primitive or thin on day 20, and thick and distinct on day 60. Representative images are shown, and panels show scale bars. Diagrams quantify size differences between time points derived from at least 300 organoids ($n = 300$) from three independent batches (**C**), VZ diameter (**D**), and CP thickness (**E**). At least 100 organoids ($n = 100$) across several batches have been sampled for size comparison. At least 25 independent brain organoids ($n = 25$) have been sampled for VZ and CP thickness. Statistical analysis was carried out by One-way ANOVA, followed by Tukey's multiple comparisons test, ***$P < 0.001$. Data presented as mean ± SEM.

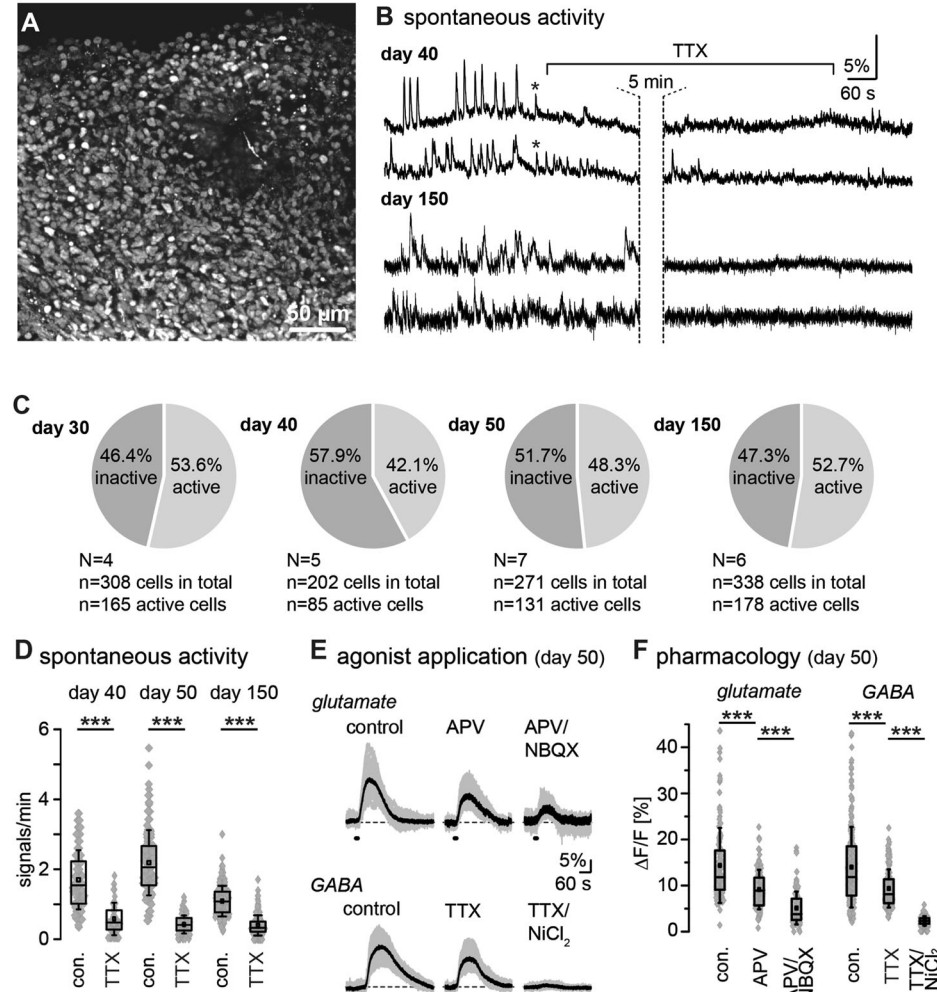

**Fig. 4 | Spontaneous and evoked activity in neural networks in Hi-Q brain organoids. A** Fluorescence intensity image of an organoid loaded with the calcium indicator OGB-1. **B** Exemplary traces of single cells revealing spontaneous calcium signaling in day 40 and day 150 old brain organoids. Asterisks highlight occasional synchronized activity. Wash-in of tetrodotoxin (TTX; 1 μM) strongly dampens spontaneous calcium signaling. **C** Pie charts illustrating active and inactive cells in brain organoids across all age groups (days 30, 40, 50, and 150). **D** Box plots showing median (line), mean (square), interquartile range (box), and standard deviation (whiskers) of the frequency of spontaneous calcium activity under control conditions and in the presence of TTX on day 40, 50, and 150 old brain organoids. Data are from three independent batches of the organoids in each age group, each containing at least four organoids ($n = 4$). Gray diamonds represent single data points/cells analyzed. **E** Intracellular calcium transients evoked by bath application of 1 mM glutamate (upper row) or 1 mM GABA (lower row) under control conditions and in the presence of ionotropic glutamate receptor inhibitors (APV for NMDARs and NBQX for AMPARs) or the presence of inhibitors of voltage- ion channels (TTX for Nav and NiCl2 for Cav) in day 50 brain organoids. Gray traces show individual cellular responses in one particular experiment; black traces are averages of the individual traces. **F** Box plots showing median (line), mean (square), interquartile range (box), and standard deviation (whiskers) glutamate- or GABA-induced calcium responses of day 50 brain organoids in control conditions and in the presence of indicated receptor/channel blockers. Data are from three independent batches of the organoids, each containing at least five organoids ($n = 5$). Gray diamonds represent single data points/cells analyzed. Gray diamonds are single data points/ cells analyzed. For (**D, E**), data are presented in Tukey box-and-whisker plots indicating median (line), mean (square), interquartile range (IQR; box), and standard deviation (whiskers). In addition, all individual data points are shown in gray underneath the Tukey plots. Data were statistically analyzed using one-way ANOVA followed by a post hoc Bonferroni test. The following symbols are used to illustrate the results of statistical tests in the figures: *: $0.01 \leq p < 0.05$; **: $0.001 \leq p < 0.01$; ***: $p < 0.001$. "n" represents the number of cells analyzed; "N" means the number of individual experiments/brain organoids. Each series of experiments was performed on at least three different organoids.

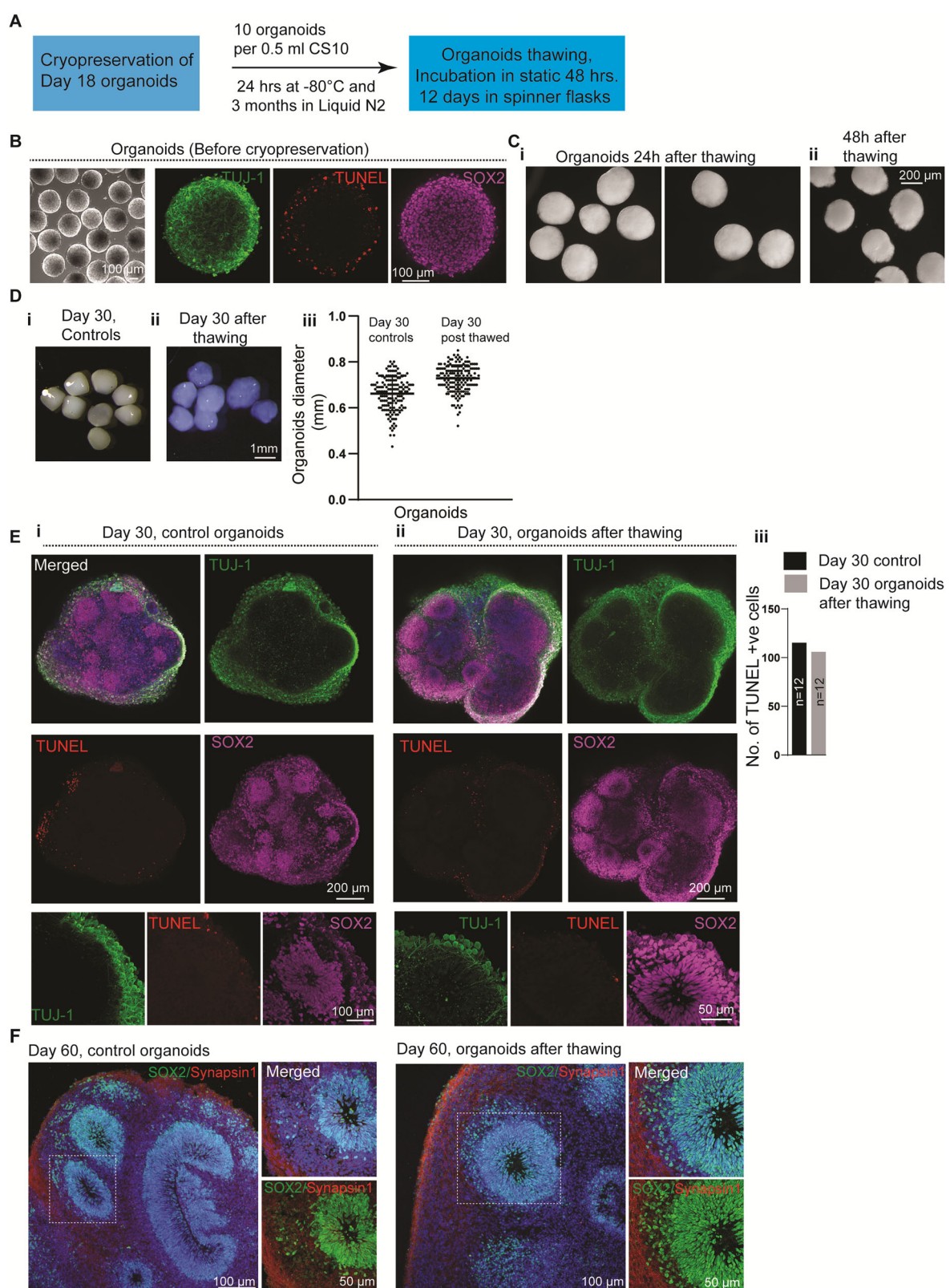

positive cells (Fig. 5E, Supplementary Fig. 6, and Supplementary Movie 15–17). By day 60, thawed and re-cultured organoids displayed synapsin-1-positive mature neuronal types and architectures indistinguishable from control organoids, which were never cryopreserved (Fig. 5F).

In contrast to 18-day-old organoids, we could not successfully re-culture day 35 or 40-old brain organoids frozen similarly.

Notably, these organoids displayed a damaged cytoarchitectural organization containing massively elevated TUNEL-positive cells, indicating that they did not recover after thawing (Supplementary Fig. 6F). These data suggest that organoids generated with the Hi-Q platform are amenable to cryopreservation at an early stage of differentiation and are viable and healthy after thawing and re-culturing.

**Fig. 5 | Cryopreservation, thawing, and re-culturing of Hi-Q brain organoids.**
**A** A schematic summarizes the cryopreservation and freeze-thawing of Hi-Q brain organoids. **B** Eighteen-day-old organoids before cryopreservation. TUJ-1 marks early neurons (green), TUNEL (red) shows the dead cells at the periphery, and SOX2 labels NPCs (magenta)—representative images from at least six experiments from three independent batches of neurospheres. Panels show the scale bar. **C** Hi-Q organoids 24 h after thawing (**i**) are still intact. After 48 h of thawing (**ii**), organoids show a slightly deformed edge. The panel shows the scale bar. **D** Day 30 Hi-Q organoids that have never been cryopreserved (**i**). The right panel (**ii**) shows a group of organoids after thawing. The color differences are due to different microscopes with filter settings. The panel shows the scale bar. (**iii**) At least eighty ($n = 80$) organoids from three independent batches ($N = 3$) were thawed and analyzed. One-way ANOVA carried out statistical analysis. There are no significant differences in the size of Hi-Q brain organoids between controls (never frozen) and freeze-thawed. Student's *t* test carried out statistical analysis. Data presented as mean ± SEM. **E, F** Comparison of the cytoarchitecture of day-30 (**E**) Hi-Q brain organoid (control, never frozen) (**i**) and frozen and thawed organoid (**ii**). Magnified panels at the bottom show a ventricular zone (VZ) with its typical cytoarchitecture of early neurons (primitive cortical plate, green marked by TUJ-1) and proliferating NPCs (magenta marked by SOX2). TUNEL (red) labels dead cells; there is no difference between control and freeze-thawed organoids ($n = 12$) across two independent batches ($N = 2$) (**iii**). Panel (**F**) shows the cytoarchitecture of 60-day-old organoids stained with SOX2 (green) and Synapsin1 (red). Panels show scale bars. At least two randomly chosen organoids across two independent batches ($N = 2$) were compared to Day 20 controls.

## Patient-derived Hi-Q brain organoids can recapitulate distinct forms of genetic brain disorders

There are more than 10,000 rare genetic diseases, and their cumulative incidence is higher than diabetes, ranging from 3.5 to 8% of the population. Notably, many of them affect brain development with poor prognoses and no cure due to the lack of mechanistic insights into disease-relevant tissue systems exhibiting various cell types and cytoarchitectures similar to the developing brain[41–43]. Since Hi-Q brain organoids possess these characteristics, we tested their versatility in modeling distinct forms of neurogenetic diseases. Firstly, we analyzed a primary microcephaly condition due to a loss-of-function homozygous mutation in a centrosome protein CDK5RAP2. To generate primary microcephaly hiPSC, we reprogrammed patient skin fibroblasts carrying a homozygous mutation in CDK5RAP2[44] (Supplementary Fig. 7A–C). Second, we modeled progeria-associated Cockayne syndrome using hiPSC derived from a patient exhibiting Cockayne syndrome (also known as Cockayne syndrome B, CSB). CSB exhibits severe neurological defects caused by a mutation in ERCC6 (also known as Cockayne syndrome B, CSB). CSB is a protein implicated in the transcription-coupled nucleotide excision repair pathway in DNA damage[45–47]. CSB-related microcephaly is a secondary microcephaly. In contrast to primary microcephaly, where the brain size is small by birth, in secondary microcephaly, the cessation of the brain growth occurs postnatally.

Although we used the same number of hiPSC (Healthy control, CDK5RAP2-mutated, and Cockayne syndrome) to generate Hi-Q brain organoids, age-matched (30-day old organoids) CDK5RAP2 patient-derived organoids (Hereafter CDK5RAP2 organoids) were smaller in size. On the other hand, organoids generated from CSB patient-derived hiPSC (hereafter CSB-Organoids) were more prominent in size which dramatically collapsed at later time points, indicating that cellular functions leading to neural epithelia formation are abnormal in both models (Fig. 6A). In addition, immunostaining for SOX2-positive progenitors and TUJ1-positive early neurons on 30-day-old Hi-Q organoids revealed that control organoids displayed structurally organized ventricular zones (VZ) abundant with progenitors in a compact organization and a layer of neurons forming a primitive cortical plate (Fig. 6Bi). In contrast, CDK5RAP2 organoids revealed structurally compromised VZs with reduced diameter, less compact progenitors' organization, and a dispersed cortical plate with TUJ1-positive neurons spreading through organoids (Fig. 6Bii).

CSB organoids (CSB 739), on the other hand, did not exhibit recognizable VZs, and the progenitors were less densely packed and randomly distributed with weakly positive SOX2 cells. Moreover, TUJ-1 positive neurons did not form a distinct cortical plate; instead, they were broadly diffused in the organoid tissue, suggesting that brain organization is perturbed in CSB organoids (Fig. 6Biii and Supplementary Fig. 7D–F). Finally, turning our analysis for spontaneous DNA-damage and apoptotic cell deaths, we noticed that CSB-organoids harbored many pH2AX-positive and TUNEL-positive cells (Fig. 6C, D). These findings revealed that without functional CSB, neural tissues undergo extensive DNA damage, thereby perturbing brain organization.

Notably, neither DNA damage nor cell death appears to cause microcephaly in CDK5RAP2 organoids, as we did not notice a significant increase in pH2AX-positive cells and TUNEL-positive cells (Fig. 6C, D). Thus, we reasoned that the microcephalic phenotype of brain organoids could be due to defective NPC maintenance, leading to premature differentiation of NPCs and NPC loss. To test this, we analyzed the kinetics of NPCs proliferation by analyzing the division plane of p-Vimentin-positive apical progenitors that form the lumen of the VZ. In contrast to controls, which harbored increased frequencies of horizontally oriented mitotic NPCs, we identified that patient-derived organoids harbored mostly vertically oriented mitotic NPCs. This indicated an unscheduled switching of the division plane, leading to premature differentiation with depletion of progenitors (Fig. 6E). This finding is similar to findings observed in EB-based organoids that have modeled microcephaly due to CDK5RAP2 mutation[9], explaining that the loss of CDK5RAP2 perturbs the horizontal orientation of the spindle in the patient tissues, which is crucial for the symmetric expansion of the NPCs pool. These comparative analyses indicate that the Hi-Q brain organoids are robust in modeling distinct neurodevelopmental disorders.

## Hi-Q brain organoids model glioma invasion and are amenable to medium-throughput compound screening

Glioblastoma (GBM) is an aggressive form of brain tumor with a poor prognosis. GBM harbor glioma stem cells (GSCs) that infiltrate the brain and account for the fatal nature of this disease for which there is no promising cure[48–50]. Clinically, the diffusive neuroinvasive behavior of GSCs in the human brain remains unpredictable[51,52]. This could be due to the lack of a physiologically relevant pre-clinical human in vitro system that can reliably recapitulate GSC invasion behaviors. Brain organoids serve as a 3D substrate for GSC and reveal the neuro-invasion behavior of GBM[34,53]. Conventionally, GSC cultures can be used for compound screening to kill or stop GSC proliferation in 2D. On the other hand, utilizing a brain tissue-like system is can identify the compound that can stop the neuroinvasive behavior of GSCs.

Therefore, we designed an assay to precisely measure the GSC invasion into 3D brain organoid tissues and identify compounds that can perturb the GSC invasion. Although brain organoids hold promise for 3D-organoid-based compound screening, it requires generating a high quantity of organoids with increased homogeneity to arrive at statistically significant results[54]. As Hi-Q brain organoids satisfy this prerequisite, we attempted to develop a proof-of-principle experiment adapting Hi-Q brain organoids for a medium-throughput drug screen.

First, we determined whether Hi-Q brain organoids can reveal the neuroinvasion behavior of patient-derived GSC. We then aimed to adapt GSC-invading organoids for a drug screening assay to identify compounds inhibiting the GSC invasion in the brain tissue. We labeled a patient-derived GSC line (#450) with mCherry and adapted our recently established brain organoid-based GBM invasion assay to Hi-Q brain organoids[33,34]. Importantly, Hi-Q brain organoids could faithfully recapitulate the invasive behavior of GSCs. In brief,

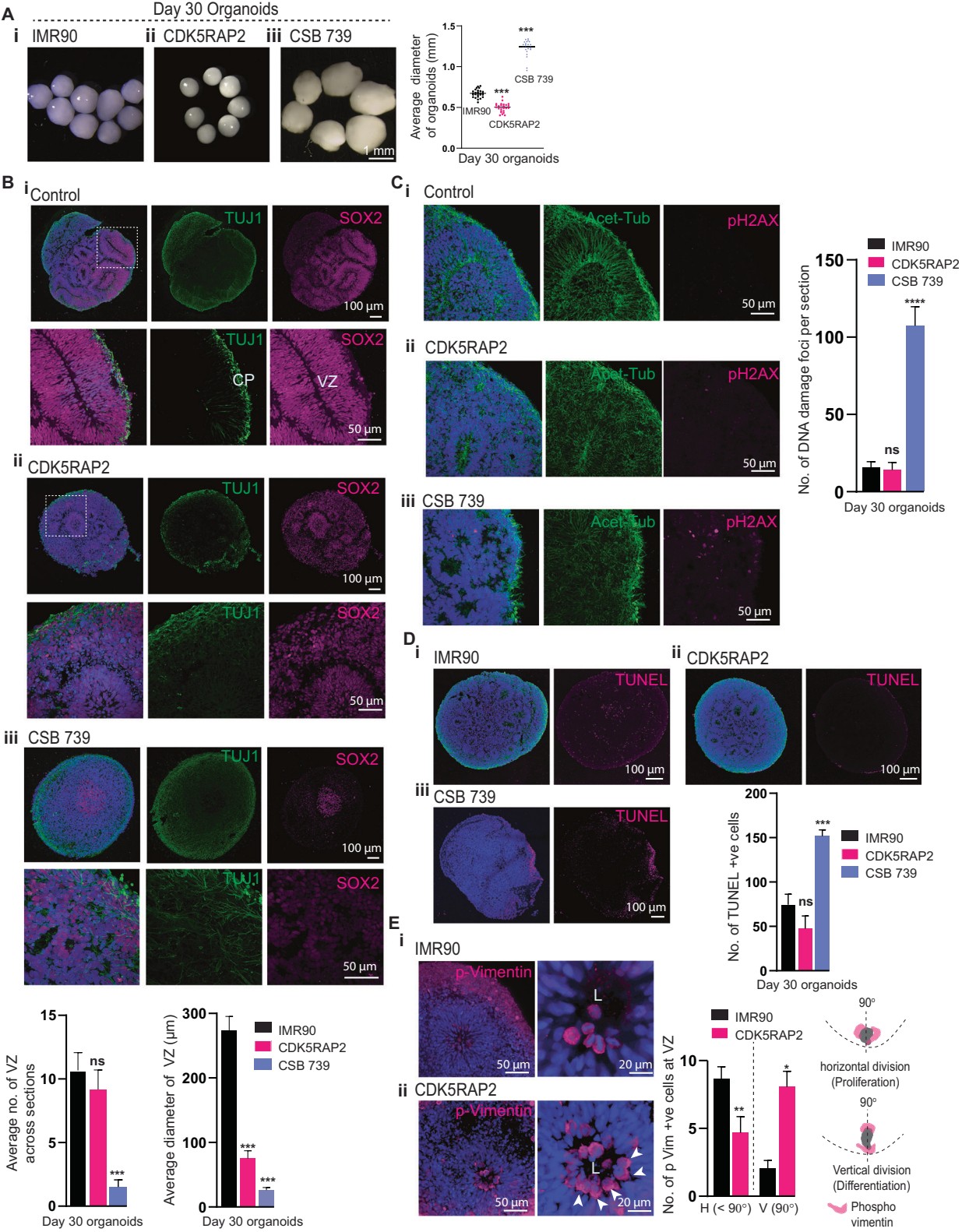

when 5000 suspended GSCs or GSC spheroids were applied to day-50 Hi-Q brain organoids, GSCs adhered to the surrounding organoids within 24 h. Within 24–72 h, GSCs infiltrated the organoids, exhibiting typical in vivo glioma invasion characteristics, such as protrusions with extended microtubes[55] (Fig. 7A, B). Using this setup, we screened compounds that could prevent the neuro-invasion property of GSCs.

We consolidated a library of 180 compounds with known biological targets to design a medium-throughput assay (Supplementary Table 3). We began the preliminary screening assay in 96-well format by incubating Hi-Q brain organoids and GSCs for 72 h with 5 μM of each compound. We imaged each invasion sample over three days with two image acquisition time points in an automated microscope programmed to acquire images at twenty-four and seventy-two hours. We used high-

**Fig. 6 | Hi-Q brain organoids model microcephaly and brain organization defects. A** Comparison of healthy control (IMR90) and Hi-Q brain organoids derived from mutant patients harboring mutations in CDK5RAP2 and Cockayne syndrome B gene (CSB 739) (**i-iii**). CDK5RAP2 brain organoids (**ii**) are microcephalic as they are significantly smaller than healthy organoids. On the other hand, CSB organoids are significantly larger than healthy controls. The panel shows the scale bar. At least twenty ($n = 20$) randomly chosen day 30 organoids were analyzed across three independent batches ($N = 3$). Statistical analysis was carried out by One-way ANOVA, followed by Tukey's multiple comparisons test, ***$P < 0.001$. Data presented as mean ± SEM. **B** Tissue sections of healthy control (IMR90), CDK5RAP2, and CSB 739 brain organoids (**i-iii**). The magnified panel under each variety shows VZ. SOX2 labels NPCs, and TUJ-1 marks primitive cortical plate (CP). Compared to healthy organoids (IMR90), CDK5RAP2 organoids have slightly disorganized and smaller VZ. Bar graphs at the bottom quantify the average number of VZs and their diameter in each kind. Panels show the scale bar. At least twenty sections ($n = 20$) from three different organoids of each group were analyzed. Statistical analysis was carried out by One-way ANOVA, followed by Tukey's multiple comparisons test, ***$P < 0.001$. Data presented as mean ± SEM. **C** CSB 739 (**iii**) but not healthy (**i**) or CDK5RAP2 (**ii**) organoids display a significantly increased level of pH2AX-positive nuclei (magenta) indicative of DNA double-stranded breaks. The bar graph at the right quantifies the relative proportions of pH2AX-positive nuclei. Panels show the

scale bar. An average of at least twenty sections ($n = 20$) from three different organoids of each group was considered. Statistical analysis was carried out by One-way ANOVA, followed by Tukey's multiple comparisons test, ***$P < 0.001$. Data presented as mean ± SEM. **D** CSB 739 (**iii**) but not healthy (**i**) or CDK5RAP2 (**ii**) organoids display a significantly increased level of TUNNEL-positive cells (magenta) indicative of dead cells. The bar graph quantifies the relative proportions of TUNNEL-positive nuclei. An average of at least twenty sections ($n = 20$) from three different organoids of each group was considered. Statistical analysis was carried out by One-way ANOVA, followed by Tukey's multiple comparisons test, ***$P < 0.001$. Data presented as mean ± SEM. Panels show the scale bar. **E** Hi-Q brain organoids reveal the kinetics of the apical progenitor division plane. P-Vimentin selectively labels the dividing apical progenitors at the VZ's apical side. Healthy brain organoids (IMR90) predominantly display dividing progenitors whose division plane is horizontal to the VZ lumen (**i**). In contrast, CDK5RAP2 brain organoids (**ii**) harbor apical progenitors whose division plane is mainly vertical to the VZ lumen. The bar diagram quantifies the distribution of the division plane. H, horizontal. V, vertical. A schematic at the right shows horizontal and vertical division planes. An average of at least fifteen sections ($n = 15$) from three different organoids of each group was considered. Statistical analysis was carried out by One-way ANOVA, followed by Tukey's multiple comparisons test, *$P < 0.1$. Data presented as mean ± SEM. Panels show the scale bar.

content image analysis to determine the invasion of GSCs into organoids. Here, spots were detected within the organoid region, and their numbers were quantified (see "Methods" section) (Fig. 7C). Via this assay, we identified sixteen compounds that negatively affected GSC invasion (Fig. 7D, E). As a secondary screen, we tested the ability of ten of these sixteen compounds (which exhibited the most inhibitory effect) to perturb GSC invasion at 1 μM concentration. This assay relied on uniform-sized organoids for comparison. This analysis identified Selumetinib and Fulvestrant as effective inhibitors of GSC invasion into brain organoids as judged by the differences in the number of GSC foci between twenty-four and seventy-two hours of invasion assay (Supplementary Fig. 8, parts 1 and 2, red dotted box in part 2). Selumetinib is a mitogen-activated protein kinase 1, and 2 inhibitors used to treat neurofibromatosis[56]. Fulvestrant is a selective estrogen receptor degrader (SERD) used to treat advanced breast cancer[57].

We then tested whether Selumetinib and Fulvestrant compounds could perturb GSC invasion when supplied as single cells or spheres in Hi-Q brain organoids (Supplementary Fig. 9, 10, Low resolution). We then applied high-resolution quantitative 3D imaging and evaluated the abilities of these compounds in perturbing GSC invasion. Compared to vehicle control, these compounds significantly prevented the invading power of GSCs when supplied with either compact spheres or single cells (Fig. 7F). To corroborate our imaging-based data and to compute the fraction of the organoid volume occupied by GSCs, we applied a computational tool (Described in the method section). We calculated the fraction of the organoid volume occupied by GSCs (Supplementary Fig. 10D).

To test whether the drugs identified in our Hi-Q brain organoid had a significant impact in vivo, we grafted GFP-tagged GSC lines onto the striatum of NOD-SCID mice (line GSC#1 and GSC#472. Figs S11). After a week of grafting, we treated the animals for three weeks with saline or a combined 20 mg/Kg of Selumetinib and Fulvestrant. In untreated mice, both GSC lines invaded extensively at the injected striatum and spread to the white matter paths, like the corpus callosum, optic tract, anterior commissure, and cerebrospinal fluid (CSF) pathways. We imaged histological sections through the tumor epicenter to assess the invasion of the white matter and CSF paths. We analyzed the number of GSCs invading the corpus callosum, optic tract, and the walls of the ventricles in saline and drug-treated subjects (Supplementary Fig. 11). The drug treatment significantly reduced the tumor volumes of GSCs in the striatum (Supplementary Fig. 11A, E, F). Besides, the drug treatment also significantly reduced the spread of tumor spheres onto the ventricular walls and impaired the invasion of GSCs into the corpus callosum and optic tract (Supplementary

Fig. 11D, E). In summary, the brain organoids generated via the Hi-Q approach are amenable to model glioma and can ultimately serve as a test system to screen potential therapeutic compounds to inhibit the invasive behavior of GSCs.

## Discussion

In vitro brain organoids have shown promise in modeling human brain development, neurological diseases, and drug screening settings. However, the organoids' utility has been limited by several shortcomings related to their heterogeneity, poor reproducibility, impaired cell diversity due to activated stress pathways, and a lack of a method to generate large number of brain organoids with reproducible qualities. Considerable efforts have been invested in developing culturing conditions to ensure high reproducibility across individual brain organoids within a batch. To this end, several works have used prefabricated multi-wells and cultured embryoid bodies, cortical spheroids, and brain organoids[7,15,16,58,59] (Summarized in Supplementary Table 4). Yet, challenges remain. Notably, a comprehensive study revealed that brain organoids generated from current methods have altered neuronal diversity due to induced stress pathways [21] and, thus, cannot reliably model diseases. Thus, no standardized method is available to fulfill these pitfalls. Yet, developing a brain organoid culturing method that can robustly model various diseases and adapt drug screening strategies is critically needed.

These aspects have prompted us to optimize culturing conditions, which led us to generate Hi-Q brain organoids. The method used a custom-made spherical plate that did not require pre-coating, centrifugation, or embedding with an extracellular matrix and generation of embryoid bodies. Notably, our protocol generates large quantities of organoids per batch exhibiting a high degree of similarities in size, shape, and cell diversities and, importantly, are free from excessive ectopic stress-inducing pathways (Fig. 2F). Although the described Hi-Q approach is better than the other in vitro methods regarding cellular stress markers[21], it still needs to meet the in vivo conditions. Nevertheless, the Hi-Q brain organoids method expressing low levels of stress markers is an improved in vitro method closer to in vivo conditions.

The Hi-Q approach is robust as we could generate organoids from at least six genetic backgrounds (Figs. 1, 6). In addition, the presence of active neuronal networks and increasing cell diversities over time indicate that Hi-Q brain organoids are physiologically relevant (Fig. 4). Furthermore, we were successful at cryopreserving day 18 brain organoids generated with the Hi-Q approach, offering the economic advantage of serving as a biobanking repository of patient-specific brain organoids (Fig. 5). Day 18 Hi-Q brain organoids are

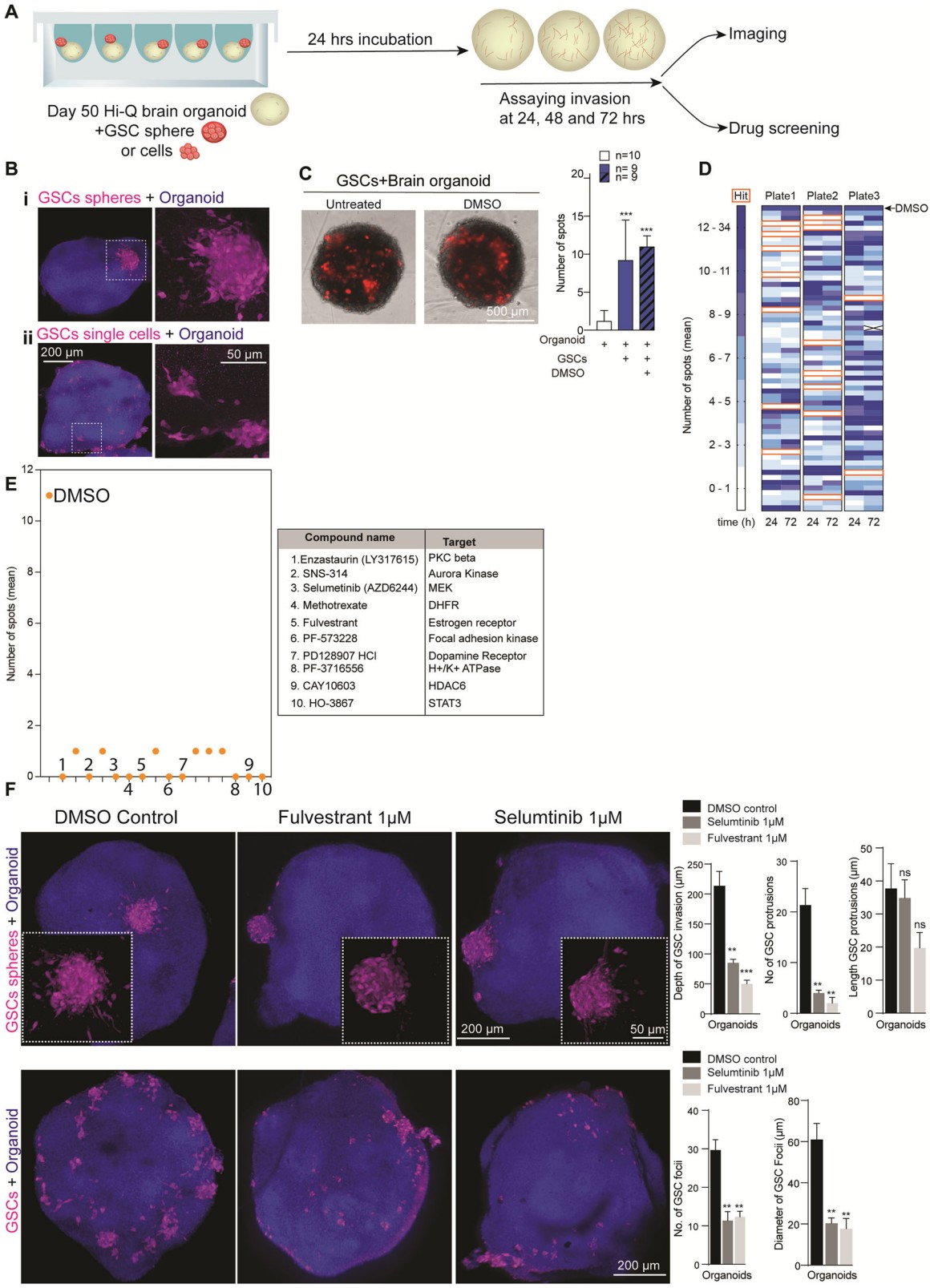

relatively immature compared to age-matched EB-based organoids and contain more proliferating cells (Supplementary Fig. 3). This could be another advantage for a better recovery after thawing the frozen organoids. We believe that freezing younger brain organoids is the first step to optimize conditions for mature brain organoids, which are abundant with post-mitotic neurons and less undifferentiated progenitors.

The Hi-Q approach significantly differs from the current methodology in at least two aspects. The first aspect is skipping the embryoid body (EB) formation. It may raise a question of how brain organoids can be generated by passing early embryonic development signaling events. In the mammalian embryo gastrulation, the ectoderm forms the outer layer and diverges into surface ectoderm and neuroectoderm. Except for the Sonic hedgehog signal from notochord to form the neural tube, no other

**Fig. 7 | Hi-Q brain organoids allow glioma invasion assays and can adapt medium-throughput drug screening assays to identify anti-glioma compounds. A** Experimental scheme for a glioma invasion assay using Hi-Q brain organoids that can be adapted for drug screening. **B** Hi-Q brain organoids allow GSC invasion as spheres (**i**) or single cells (**ii**). The magnified panel at the right shows invading GSCs with their typical protrusions into the organoid tissues (arrows). The bar diagram at the right quantifies the depth invaded by GSCs. Shown are representative images from at least six experiments from three independent batches of organoids. Panels show the scale bar. **C, D** Assay design adapted to screen the compounds preventing GSC invasion. Representative images showing invading GSC into Hi-Q brain organoids (**C**). At least 12 organoids ($n = 12$) from three independent experiments were analyzed. Statistical analysis was carried out by One-way ANOVA, followed by Tukey's multiple comparisons test, ***$P < 0.001$. Data presented as mean ± SEM. Automated imaging and computer-based counting of GSC spots in each organoid. Two organoids were tested for each compound. The algorithm assigns a space when there are no invading GSCs. Dark blue denotes

more GSC spots in organoids. Each box represents a well that is supplied with a compound (**D**). **E** Dot plot shows the selected compounds that prevent GSC invasion into organoids. Each dot denotes the number of GSC spots in organoids. The table at the right lists selected ten compounds that can stop the GSCs' invasions into brain organoids. The biological functions of each compound are also listed. **F** Quantitative 3D imaging of GSCs invasion into brain organoids. The top panel shows the GSC invasion from spheres. The bottom panel shows GSCs invasions from GSCs suspension. Fulvestrant and Selumtinib significantly prevent GSC invasion into brain organoids as selected compounds. The invaded GSC (as spheres or single cells) were marked with red arrowheads. White arrowheads point to the spheres or cells that fail to invade the organoids in the presence of drugs. Panels show the scale bar. The bar diagram quantifies the inhibitory effects of the selected compounds on GSC invasion. Average measurements from at least ten organoids ($n = 10$) per drug condition were considered. Statistical analysis was carried out by One-way ANOVA, followed by Tukey's multiple comparisons test, ***$P < 0.001$, **$P < 0.01$. Data presented as mean ± SEM.

signals from mesodermal or endodermal structures have been described for the ectodermal development[60]. Indeed, there is evidence for crosstalk between the surface and neuroectoderm[61]. Thus, our results show that skipping the EB step in brain organoid generation may be advantageous as it could favor cells to differentiate along the ectodermal lineage. First, irregular sizes of EBs and their quality can cause heterogeneous brain organoids[18–20]. Second, extensive development of meso- and endodermal lineages from EBs might influence the differentiation of the ectodermal cells that cannot be reversed by the commonly used compounds such as retinoic acid, Noggin, or SB431542. Finally, by directly inducing differentiation into neural epithelium from hiPSC, we could generate brain organoids with enriched neuro- and surface ectoderm-derived cell types. Notably, these brain organoids expressed various cell types, electrically active neural networks, and optic vesicles[3,26,27].

In the second aspect, we differentiate hiPSC directly into neurospheres using a micropatterned plate. This allowed us to perform the initial differentiation steps of hiPSC in Matrigel and ROCK inhibitor-free conditions and obtain uniform-sized neurospheres, the early neural tissue intermediates. Finally, the entire protocol follows unguided differentiation and requires minimal manual handling of cells. As the method already regulates the uniform-sized neurospheres, aberrant growth of organoids with varying sizes at later stages is limited.

Our comparative sc-RNA analysis shows that Hi-Q brain organoids are relatively less mature than EB-based brain organoids, which contain more differentiated neurons and astrocytes[7,32] (Supplementary Fig. 3). The following reasons could reconcile the differences in the maturation state of Hi-Q and methods that use EB-based brain organoids. Our method omits EB formation by directly exposing hiPSC to neural induction media (NIM). At this early stage, NIM does not trigger neuronal differentiation but allows the generation of pure neuroectoderm to form neural epithelia containing neural progenitors. The idea behind this strategy is to get more homogeneous neural lineages. At the differentiation step, our method does not use retinoic acid or any neuronal maturation factor, such as BDNF, which has been shown to promote neuronal differentiation[8,62]. This could explain why we observe less matured neuronal populations in Hi-Q organoids than in the methods that have used EB-based organoids. In other words, the Hi-Q method does not affect neuron diversity but may allow controlled differentiation. While our method indicates that neuroectoderm formation does not require meso- and ectoderms, we cannot exclude the possibility that these two germ layers impact neuronal differentiation and maturation. We also believe one can optionally add factors that can enhance neural differentiation or maturation, such as retinoic acid, Brain-derived neurotrophic factor (BDNF), and Ciliary neurotrophic factor (CNTF) at any time of the culture.

The described Hi-Q brain organoids show reduced ectopic stress (Fig. 2F), possibly due to several factors combined. First, the Hi-Q method minimizes manual handling of hiPSC, such as embedding

them with Matrigel and incubating them in various dishes before transferring them to spinner flasks. Second, the described method does not use the ROCK inhibitor throughout the culturing. Third, the custom-made plate did not require precoating or centrifugation of cells, which will avoid stress due to gravitational force.

3D brain organoids recapitulate human brain development principles and thereby hold promise to decipher genetic brain diseases that pose a socioeconomic burden. Notably, there are more than 300 rare genetic brain diseases. When combined, which is not rare. Economically, it is challenging to generate rodent models for each of these diseases. Even developed rodent models may not help because human brain development differs fundamentally from rodent brains[63,64]. Hi-Q brain organoid platform could solve this urgent need by modeling rare diseases in a high throughput fashion and identifying shared pathways among disease groups and therapeutic targets. Our developed application in this work indeed tests the ability of Hi-Q brain organoids to model at least two different types of genetic brain diseases. When modeling CDK5RAP2-mutated microcephaly, Hi-Q brain organoids could recapitulate the premature differentiation of NPCs, a mechanism attributed to causing the depletion of NPCs causing microcephaly (Fig. 6). Conversely, when modeling progeria-related microcephaly due to mutations in CSB, the Hi-Q brain organoids displayed distinct phenotypes exhibiting DNA damage and defects in brain organization.

Another developed application in this work uses Hi-Q brain organoids to study GBM invasion behaviors (Fig. 7). The conventional preclinical model to assay the GBM invasion is grafting the human GBM sample into the mouse brain[65,66], which has limitations such as its lengthy process and lack of a tumor microenvironment similar to the human brain[51]. Our proof-of-principle experiment in this work has successfully used Hi-Q brain organoids in a medium-sized drug screening assay and identified two compounds that effectively perturbed the invasion behavior of patient-derived glioma stem cells (GSCs) in vitro and mouse xenografts.

In summary, our Hi-Q technology solves many of the limitations in brain organoid research. By generating brain organoids in large quantities in a versatile and robust way, we trust that our approach and developed applications will pave the path for personalized medicine through disease modeling and high throughput compound screening.

## Methods

All hiPSC cell lines were maintained in mTeSR1 medium on Matrigel-coated tissue-culture dishes (Supplementary Table 5). Human hiPSC were cultured in mTeSR1 medium (Stem cell technologies) on Matrigel (Corning) coated cell culture dishes at 37 °C and 5% $CO_2$ and routinely checked for mycoplasma contamination with the MycoAlert Kit (Lonza). Media change was carried out once in 2 days. The cells were split 1:4 onto fresh Matrigel-coated dishes using an enzyme-free method, employing ReLSR (Stemcell Technologies, Catalog # 05872) as per the manufacturer's protocol. Studies with hiPSC, patient-

derived hiPSC (CSB-GM739 and CDK5RAP2) included informed consent from the patient's guardians. Patient-derived brain organoids have been conducted according to institutional regulations and were approved by the ethics committee of the Heinrich-Heine-University, Düsseldorf (Study number 2018-272).

## Generation of Hi-Q brain organoids

**Seeding hiPSC and neural induction.** hiPSC (at 80% confluency) were dissociated into single cells by treatment with Accutase (Sigma-Aldrich) at 37 °C for 5 min. The cells were then centrifuged at $100 \times g$ for 3 min and resuspended in 1 ml of STEM Diff Neural induction medium (NIM) (Stem cell technologies, Catalog # 05835), supplemented with 10 µM ROCK inhibitor (Selleckchem, Cat # S1049). The cells were counted using a hemocytometer, and 3 x $10^6$ cells were suspended in 1 ml of NIM medium, supplemented with 10 µM ROCK inhibitor. The resuspended cells were carefully added into our custom-made spherical microwell plate containing 1 ml of NIM, reaching a final volume of 2 ml in each well of the microwell plate. The cell suspension was carefully mixed by pipetting 5-6 times to distribute the cells evenly. When a commercial plate is used (AggreeWell 800), the plates were sealed with parafilm, and was centrifuged at $300 \times g$ for 8 min. The plate was then turned and centrifuged again to ensure the cell seeding at the center of the microwells, assuming each microwell receives approximately 10,000 cells. The parafilm was removed, and the plate was incubated at 37 °C and 5% $CO_2$ overnight.

**Neurosphere formation and transfer to spinner flasks.** For the next five days, partial media change was done by replenishing 0.8 ml of NIM with 1 ml of NIM daily without the ROCK inhibitor. During this period, the hiPSC formed strong interconnections, which gave them a 3D appearance known as neurospheres. After 5 days of partial medium change, the neurospheres were detached from the microwells by pipetting them slowly with DMEM/F12 medium (Gibco). Next, these neurospheres were transferred with a cut 1000 µl pipette tip into a 40 µm cell strainer (Corning, Cat # CLS431750). This procedure was repeated until all the neurospheres were detached and carefully transferred. The cell strainer was then inverted, and the neurospheres were released into a 10 cm cell suspension TC- dish using approximately 5-10 ml of the Neurosphere Medium consisting of 1:1 DMEM/F12 and Neural Basal medium, supplemented with N2 (1:200), B27 w/o Vitamin A (1:100) (Thermo-scientific) 0.05 mM MEM non-essential amino acids (Gibco), L-glutamine (1:100, Gibco), Pen-strep (100 µg/ml each), Insulin (0.2755 µM, Sigma Aldrich), 0.05 mM β-Mercaptoethanol (Life Technologies).

Once all the neurospheres were released, they were transferred into 500 ml spinner flasks (Integra biosciences, Cat # 10610762) containing 75 ml of the Neurosphere organoid medium. Optionally, (if the neurospheres look deformed) the released neurospheres could be incubated TC-dish for two to four days stationarily before being into spinner flasks. The flasks were then incubated at 37 °C on the magnetic platform. The neurospheres were subjected to a rotation setting at a constant speed of 25 rpm. Half media change was performed once every three days. After four days, the neurosphere medium was switched to the brain organoid differentiation medium.

After 21 days, the organoids were further cultured in the Human brain organoid maturation medium devoid of the SMAD inhibitors (SB431542 and Dorsomorphin). The brain organoids were allowed to mature and analyzed at different time points for maturation markers. Supplementary Table 1 lists media composition and incubation duration.

## Time-resolved scRNA-seq of Hi-Q organoids

For scRNA library construction, we used the Chromium™ Single Cell 3' Reagent Kits v3. Single nucleus suspensions in 1XPBS containing 0,04% BSA (700-1200 nuclei/µL concentration) have been checked for viability, debris, and cell aggregates. To achieve single-cell resolution, the cells are delivered at a limiting dilution, such that the majority (-90-99%) of

generated GEMs contain no cell. In contrast, the remainder essentially includes a single cell. Because of the complex composition of organoids, we aim to target 2.000-3.000 cells per sample. Upon dissolution of the Single Cell 3' Gel Bead in a GEM, primers containing (i) an Illumina R1 sequence (read 1 sequencing primer), (ii) a 16 bp 10x Barcode, (iii) a 12 bp Unique Molecular Identifier (UMI) and (iv) a poly-dT primer sequence are released and mixed with cell lysate and Master Mix. Incubation of the GEMs then produces barcoded, full-length cDNA from poly-adenylated mRNA. After incubation, the GEMs are broken, and the pooled fractions are recovered. Silane magnetic beads remove excess biochemical reagents and primers from the post-GEM reaction mixture. Full-length, barcoded cDNA is then amplified by PCR to generate sufficient mass for library construction.

Enzymatic Fragmentation and Size Selection optimize the cDNA amplicon size before library construction. R1 (read 1 primer sequence) is added to the molecules during GEM incubation. P5, P7, a sample index, and R2 (read 2 primer sequence) are added during library construction via End Repair, A-tailing, Adaptor Ligation, and PCR. The final libraries contain the P5 and P7 primers used in Illumina bridge amplification. A Single Cell 3' Library comprises standard Illumina paired-end constructs that begin and end with P5 and P7. We allocated Illumina NovaSeq6000 flowcells to sequence with the first read 28nt (cell-specific barcode and UMI) and generate with the second read 90nt 3´mRNA transcriptome data. Using the v3 chemistry version, 25kreads/nucleus are sufficient for comprehensive scRNA analysis. Single-cell RNA-seq of five batches of organoids was performed using the 10× Chromium pipeline.

Reads were mapped to the human genome (GRCh38), and count matrices were generated using CellRanger. Python notebooks were used to analyze the count matrices by custom functions and the package Scanpy v1.8.2 (Wolf, Angerer and Theis, 2018). Raw single-cell count matrices have been produced for each batch of organoids at the three time points. Batches of the same time point have been concatenated, and possible doublets have been filtered out using the scrublet package (Wolock, Lopez, and Klein, 2019). The three matrices have been then concatenated and filtered. Cells were kept if expressing between 200 and 5000 genes, having between 1000 and 40000 counts, and showing lower percent mitochondrial counts than 8%. Counts have been normalized and log-transformed.

Then, the set of highest variable genes has been detected. The total number of counts per cell and the percentage of mitochondrial counts have been regressed in the data. The PCA dimensions have been calculated, and the harmony package has smoothed the batch effect between time points[67]. Additionally, the $k$-nearest neighbor was computed by the bbknn algorithm[68]. The 2-dimension representation of such a network has been calculated as a force-directed graph by the Force Atlas2 algorithm[69]. A Lasso regression model has been trained on cell type labels of a published dataset of embryonic human brain (Nowakowski et al. [31]). Such a model has been used to score labels on each cell in the Hi-Q dataset. We assigned to each cell the label with the higher prediction score obtained. Such cell-type labels have been used to compare batches between the same and different time points. The proportions of each cell type label served as features for principal coordinate analysis to determine distances between samples. The Leiden algorithm and marker genes have calculated unbiased clusters by t-test. The pseudotime ordering was computed using the diffusion pseudotime (DPT) algorithm by setting the proliferative radial glia cells as roots[70]. To compare sc-RNA data between Hi-Q and EB-based brain organoids, we processed the data for integration using the Seurat (Ver.5) and Harmony (Ver 1.2) packages.

## Hi-Q brain organoid fixation, permeabilization, antigen retrieval, and blocking

Whole brain organoids were fixed for 1 hr using 4% Paraformaldehyde at 37 °C. The organoids were then washed with 1x PBS containing 30 mM glycine (2–3 times), permeabilized with 1x PBS containing 0.5%

Triton X 100, 0.1% Tween 20 for 15 min, and further blocked with 1x PBS containing 0.5% Fish Gelatin for 1 h at room temperature (RT). The organoids were treated with 1% SDS for 5 min for nuclear staining and washed 3x times with 1x PBS. The organoids were then blocked, as described previously.

Brain organoids were first fixed with 4% paraformaldehyde for 20 min for organoid tissue sections and then suspended in 30% sucrose overnight at 4 °C, allowing them to sink. The following day, the organoids were embedded in a tissue-embedding solution containing (7.5% gelatin and 10% sucrose in PBS)[71]. Next, the embedded organoids were placed on dry ice for rapid freezing and transferred to –80 °C overnight. Finally, the frozen organoids were subjected to cryo-sectioning to obtain 20 μm slices using Cryostat Leica CM3050 S.

### Immunofluorescent staining

The Hi-Q organoids (whole mounts or tissue sections) were fixed and blocked as described above. The organoids were then moved to 2.0 ml Eppendorf tubes for immunostaining for whole mounts. Mouse anti-Nestin (1:100, Novus Biologicals), mouse anti-SOX2 (1:50, Abcam), rabbit anti-Arl13b (1:100, Proteintech), rabbit anti-Doublecortin (1:100, Synaptic Systems), rabbit anti-MAP2 (1:200, Proteintech), rabbit anti-synapsin-1 (1:200, Cell Signaling),Rabbit anti-TUJ1(1:400, Sigma - Aldrich), Mouse anti-acetylated tubulin (1:400, Sigma-Aldrich), Mouse anti-Tau (1:100, DSHB), Rat anti-CTIP2 (1:300, Abcam), Rabbit anti-PCP4 (1:100, Proteintech), Rabbit anti-pH2AX (1:400, Cell signaling), Mouse anti-Actin (1:100, R&D systems), rabbit anti-PSD95 (1:100, Proteintech), mouse anti-PAX6 (1:100, Proteintech), mouse anti-p-vimentin (1:200, Abcam) and TUNEL staining kit (Thermo Fischer). We used Alexa Fluor Dyes conjugated with goat/donkey anti-mouse, anti-rabbit, or anti-rat (1:1000, molecular probes, Thermo Fisher, USA) for secondary antibodies. In addition, DAPI 1 μg/ml (Sigma Aldrich, USA) was used to stain the nucleus.

### Whole organoid tissue clearing and mounting

Hi-Q organoids were tissue-cleared after immunostaining using increasing concentrations of ethanol (50%, 70%, and 100%) (PMID: 32521263). The organoids were treated with different percentages of ethanol for approximately 5–6 min at RT with constant mixing. After treatment with 100% ethanol, organoids were removed and allowed to dry till all the ethanol evaporated. Next, the organoids were cleared by adding 100 μl of ethyl cinnamate (Sigma-Aldrich, Cat. 8.00238) for 10 min. Once cleared, the organoids were transferred with a cut 1000 μl pipette tip to the μ-slide angiogenesis chamber (ibid, Cat # 81506), which could readily be imaged.

### Confocal microscopy

Tissue sections or whole-mount tissue-cleared organoids were imaged using a Zeiss LSM 880 confocal microscope. Our experiments used 405, 488, 561, or 633 nm laser lines and were imaged using objective 10x/0.3, 20x/0.8 M27, and 40x/1.3 oil objectives. After tissue clearing, the whole-brain organoids were suspended in ethyl cinnamate and imaged in the μ-slide angiogenesis chamber. Z-series were obtained for both whole organoids and organoid slices, and the slice interval between the stacks was approximately 1–2 μm. The raw images acquired were processed using ImageJ, Adobe Photoshop 2020, and Adobe Illustrator 2020. Movies of organoid Z-stack images were processed using ImageJ.

### Statistical analysis

The statistical analyses were performed using GraphPad Prism (version 9). Most experiments were carried out in triplicates and analyzed using non-parametric one-way ANOVA or Student's $t$ test. The Post hoc test included primarily Tukey's Test. All values were expressed as mean ± sem. 'N' represents the number of organoids or components, and 'n' represents the number of experimental replicates. Statistical significance is represented as ns-nonsignificant, $*p < 0.1$, $**p < 0.01$, $***p < 0.001$.

### Calcium imaging

Brain organoids were cut into two halves, put with the cut surface down onto an experimental chamber, and fixed with a grid. Preparations were perfused with artificial cerebrospinal fluid (ACSF) at room temperature (20–22 °C), containing (in mM): 125 NaCl, 2.5 KCl, 2 CaCl2, 1 MgCl2, 1.25 NaH2PO4, 26 NaHCO3, and 20 glucose, bubbled with 95% O2 and 5% CO2, resulting in a pH 7.4 and an osmolarity of 310 ± 5 mOsm/l (all chemicals purchased from Sigma-Aldrich (Munich, Germany), except for indicator dyes and tetrodotoxin (TTX; BioTrend Chemicals AG, Zürich, Switzerland)). Experiments were performed at room temperature as well. For imaging of intracellular Ca2 +, the membrane-permeable form of the chemical Ca2+ indicator dye Oregon Green BAPTA 1 (OGB1-AM; Thermo-Fisher Scientific, Waltham, USA) was dissolved in HEPES-buffered saline composed of (in mM) 125 NaCl, 2.5 KCl, 2 CaCl2, 2 MgCl2, 1.25 NaH2PO4 and 25 HEPES, pH 7.4 (adjusted with NaOH) OGB1-AM was then bolus-injected (5 s/2-3 PSI) into organoid preparations at a depth of about 30-50 μm, this injection was repeated at up to 10 adjacent positions in the field of view. Imaging experiments were commenced 30 min after dye injection, allowing for de-esterification of OGB1-AM.

Wide-field Ca2+ imaging was performed using a digital variable scan system (Nikon NIS-Elements v4.3, Nikon GmbH Europe, Düsseldorf, Germany). It was equipped with a 40x/N.A. 0.8 LUMPlanFI water immersion objective (Olympus Deutschland GmbH, Hamburg, Germany) and an orca FLASH V2 camera (Hamamatsu Photonics Deutschland GmbH, Herrsching, Germany). OGB1 was excited at 488 nm, and emission was collected >500 nm at 14-20 Hz. Fluorescence emission was recorded from defined regions of interest (ROI) representing cell bodies employing NIS-Elements software (Nikon GmbH Europe, Düsseldorf, Germany). For background correction, an ROI in the field of view apparently devoid of cellular structures was chosen, and its fluorescence emission was subtracted from that derived from each cellular ROI. Background-corrected signals were analyzed offline using OriginPro Software (OriginLab Corporation v9.0, Northampton, USA). Changes in intracellular Ca2+ were expressed as ΔF/F0, representing changes in fluorescence over time (ΔF) divided by the average baseline of each ROI (F0). Only events with peak amplitudes higher than two times the standard deviation of the baseline noise were analyzed. Glutamate or GABA was applied by perfusing preparations with ACSF, to which the neurotransmitters were added at a concentration of 1 mM. Pharmacological antagonists were dissolved in aCSF and washed 15 min before the experiment, allowing sufficient blocking of the target receptor or channel.

After wide-field Ca2+ imaging, some preparations were transferred to an upright confocal laser scanning microscope (Nikon C1 Eclipse E600FN, Nikon GmbH Europe, Duesseldorf, Germany) with a 40x/N.A. 0.8 LUMPlanFI water immersion objective (Nikon GmbH Europe, Düsseldorf, Germany) and a 488 nm argon laser (Melles Griot GmbH, Bensheim, Germany). Z-stacks of the imaged region were generated (up to 240 images at a step size of 0.1 μm). They were post-processed, employing deconvolution software to delineate cellular morphology (Huygens Professional, SVI imaging, Hilversum, Netherlands).

Unless otherwise specified, data are presented in Tukey box-and-whisker plots indicating median (line), mean (square), interquartile range (IQR; box), and standard deviation (whiskers). In addition, all individual data points are shown in gray underneath the Tukey plots. Data were statistically analyzed by one-way ANOVA followed by post hoc Bonferroni test. The following symbols are used to illustrate the results of statistical tests in the figures: *: $0.01 \leq p < 0.05$; **: $0.001 \leq p < 0.01$; ***: $p < 0.001$. "n" represents the number of cells analyzed; "N" represents the number of individual experiments/brain organoids. Each series of experiments was performed on at least three different organoids.

## Data analysis and statistics (related to calcium imaging experiments)

Unless otherwise specified, data are presented in Tukey box-and-whisker plots indicating median (line), mean (square), interquartile range (IQR; box), and standard deviation (whiskers). In addition, all individual data points are shown in gray underneath the Tukey plots. Data were statistically analyzed by one-way ANOVA followed by post hoc Bonferroni test. The following symbols are used to illustrate the results of statistical tests in the figures: *: $0.01 \leq p < 0.05$; **: $0.001 \leq p < 0.01$; ***: $p < 0.001$. "n" represents the number of cells analyzed; "N" represents the number of individual experiments/brain organoids. Each series of experiments was performed on at least three different organoids.

## Cryopreservation, thawing, and re-culturing of Hi-Q brain organoids

The required number of immature brain organoids on day 8 was carefully collected from the spinner flasks and placed in a 6 cm petri dish containing a 5 ml organoid medium. With the help of a cut 1000 μm filter tip, these brain organoids were transferred into freezing vials containing commercial freezing media (CS10, Stemcell Technologies). Approximately 10 brain organoids were frozen per vial containing 1 ml of CS10. The freezing mixture containing the organoids was kept on ice for 5 min before being transferred to an isopropanol chamber at −80 °C for controlled freezing. In total, 24–48 h later, these frozen vials were transferred to liquid nitrogen tanks. The frozen organoids were re-thawed after 5–7 days to check for organoid viability, morphology, and functionality. The frozen organoids were rapidly thawed at 37 °C and then transferred with a cut 1 ml sterile pipette tip into a 6 cm dish containing 5 ml organoid medium. These organoids were placed at 37 °C in static conditions for 48 h to recover from thawing. On Day 3, the organoids were carefully transferred to a spinner flask containing 75 ml of brain organoid medium and allowed to mature in the stirred conditions described above.

## Glioblastoma invasion assays, quantification and drug screening in Hi-Q organoids

Glioblastoma primary cell line 450-cherry was maintained as described previously[33,34]. Invasion assays were performed by single-cell inoculation or by fusion of GSC spheres with our brain organoids. Plate and liquid handling were performed using a high-throughput screening workstation for organoid screening. Three plates of the Target Selective Screening Library (Selleckchem) were used for the screening (180 validated selective inhibitors dissolved in DMSO, 1 mM stock solution). Hi-Q organoids (untagged hiPSC cells with GBM-Cherry) were cultivated in CellCarrier Spheroid ULA 96-well Microplates (PerkinElmer). 72 h after organoids were placed in the plates, they were either treated with a 5 μM compound (dissolved in DMSO) or alone. In parallel, one plate was treated with control compounds. The final DMSO volume concentration was kept below 0.5%. Organoids were cultivated at 37 °C with 5% CO2 and imaged after 24 h and 72 h. Imaging of two channels (Brightfield and DsRed) was performed with the automated Operetta® High-Content microscope (Perkin Elmer)–2x objective, 1 field per well, 20 planes.

Image analysis was performed using the Columbus software (PerkinElmer). The following analysis steps in Columbus are described: as an input image, a maximum projection of the different planes was used. Brightfield signal was inverted via the 'Filter image' tool and then used to detect the organoid (Find Nuclei, Method: B, Common Threshold: 0.4, Area: >5000 μm², Splitting Coefficient: 7, Individual Threshold: 0.4, Contrast: >0.1). The DsRed channel was smoothened using 'Mean Filter' and then used to detect invasion of glioma stem cells via the 'Find spot' tool. (Method: A, Relative Spot Intensity: > 0.08, Splitting Sensitivity: 0.9, Calculate Spot Properties). Compounds were defined as hit compounds if the compound showed only 0 or 1 spot in both time points, i.e., 24 h and 72 h. To quantify invasiveness into organoids, we binarized the image of organoids invading GSC spheres using intensity-based thresholding. The noise was removed from binarized images with median filtering, and holes were filled to generate continuous masks for invading cells and the organoid. Finally, as a metric capturing the extent of invasiveness, we computed the fraction of the organoid volume containing tumor cells.

## Intracranial grafting of GSCs in immunocompromised mice

Animal experiments were performed per institutional regulations and were approved by the Ethics Committee of the Istituto Superiore di Sanità, Rome (Pr. No. 4701/17). Immunosuppressed NOD-SCID mice (male; 4–6 weeks old; Charles River, Italy) were anesthetized with an intraperitoneal injection of diazepam (2 mg/100 g) followed by intramuscular injection of ketamine (4 mg/100 g) and immobilized in a stereotactic head frame. A 2 mm right of the midline and 1 mm anterior to the coronal suture hole was made in the skull where the cells (2 × 10^5 GFP-expressing GSCs of either GSC#1 line or GSC#472 line) in 5 μl DMEM were slowly injected with a 10-μl Hamilton micro syringe, placed at a depth of 3 mm from the dura. One week later (BBS, the blood-brain barrier was assumed to still be disrupted at the injection site, the mice were assigned into four groups (I) either injected with GSC #1 and treated one time per day for three weeks with Saline (2 ml) subcutaneously (sc) or (II) Selumetinib (20 mg/kg; oral gavage) and Fulvestrant (1 mg; sc) or (III) either with GSC#472 cell line and Saline /2 ml; sc) or (IV) Selumetinib (20 mg/kg; oral gavage) and Fulvestrant (1 mg; sc). Body weight and neurological status were monitored daily. The number of animals used for each condition was (I) $n = 4$; (II) $n = 6$; (III) $n = 4$ and (IV) $n = 6$.

Eight weeks after cell engraftment, mice were deeply anesthetized and transcardially perfused with PBS (pH 7.4) and then fixed with 4% Paraformaldehyde (PFA; in PBS). The brain was prepared and dehydrated in 30% sucrose overnight (4 °C). The 40 μm serial sections (coronal plane) were done with the cryotome. Cryosections were mounted on slides and mounted with Eukitt. High-resolution imaging was done with a Laser Scanning Confocal Microscope (IX81, Olympus Inc., Melville, NY, USA). GSC invasion in the cranio-caudal extension of the brain was assessed on serial coronal sections. Digitized sections (320 μm apart from each other) were made to evaluate tumor volume.

The brain region containing GSCs was demarcated on each image, and the area was calculated. Then, each area of the infiltrated brain was multiplied for the distance to the consecutive digitized section (starting from the tumor epicenter to the cranial and caudal poles of the tumor), and partial volume values were summed up.

## Reporting summary

Further information on research design is available in the Nature Portfolio Reporting Summary linked to this article.

# Data availability

All reagents generated in this study are available from the lead contact with a completed Materials Transfer Agreement. Single-cell RNA-seq raw data that support the findings of this study have been deposited at SRA number PRJNA1091065, and gene expression counts file along with long meta-data and code notebook is deposited with DOI at https://doi.org/10.5281/zenodo.14102319. Further information and requests for resources and reagents should be directed to and fulfilled by the lead contact, Jay Gopalakrishnan jay.gopalakrishnan@uni-jena.de. Source data are provided in this paper. Source data are provided with this paper.

## Code availability

The original code for scRNA analysis of Hi-Q brain organoids has been deposited and is publicly available at https://github.com/Gpasquini/HiQ_analysis_reproducibility and Zenodo, with DOIs at https://doi.org/10.5281/zenodo.14102319.

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

## Acknowledgements

We thank the members of the Laboratory of Centrosome and Cytoskeleton Biology (CCB). J.G. acknowledges support from the Deutsche Krebshilfe (70114276), Fritz Thyssen Foundation (10.20.2.031MN), Innovation funding from BMBF VIP+ (03VP10540), and Deutsche Forschungsgemeinschaft (DFG, German Research Foundation)—Project-ID 503306912—FOR5547; DFG, GO 2301/5-2, SPP2127; J.G., N.J.Y. and L.F. acknowledge the support Glioma-PerMed consortium under ERA-NET co-fund scheme (NJY: The Research Council of Norway Grant 342432); R.P. is supported by AIRC (IG 2019 Id.23154); V.B. is funded by Volkswagen Foundation (Freigeist—A110720) and the Deutsche Forschungsgemeinschaft (BU 2974/3-2 - SPP2127, EXC-2151-390873048-Cluster of Excellence— ImmunoSensation2 at the University of Bonn). CRR: supported by the Deutsche Forschungsgemeinschaft, RU 2795 ("Synapses under Stress"), Ro2327/13-2.

## Author contributions

A.R., J.G., E.G., and K.H. conceived the concept, and A.R. performed most of the experiments; G.P., V.B., and V.J. conducted sc-RNA analysis; N.G. and C.R. conducted calcium imaging; O.S.V. and N.A. performed cryofreezing and thawing; H.W. and S.M. performed immunostaining: I.R., K.H., and S.L. involved in assay designing; A.M. and D.R. involved in imaging; A.M. involved in plate designing; O.G. contributed in iPS cells; L.R.V., Q.G., and R.P. conducted mouse xenograft experiments, B.W. and A.M. offered cell lines; L.F. processed the images; N.J.Y. processed images analyzed data.

## Funding

## Competing interests

The authors (A.R., J.G., and E.G.) have filed an international patent related to the methodology and applications (Ref: PCT/EP2021/080414). J.G. and E.G. are co-founders of NeuronFab GmbH. All other authors declare no competing interests.

## Additional information

[1]Institute of Human Genetics, University Hospital, Friedrich-Schiller-Universität Jena, 07740 Jena, Germany. [2]Department of Ophthalmology, University Hospital Bonn, Medical Faculty, Bonn, Germany. [3]Institute of Neurobiology, Faculty of Mathematics and Natural Sciences, Heinrich-Heine-Universität, 40225 Düsseldorf, Germany. [4]Research Unit Signaling and Translation, Helmholtz Zentrum München, 85764 Neuherberg, Germany. [5]Kugelmeiers Ltd, Erlenbach, Switzerland. [6]Institut de la Vision, Sorbonne Université, INSERM, CNRS, F-75012 Paris, France. [7]Department of Oncology and Molecular Medicine, Istituto Superiore di Sanità, Viale Regina Elena 299, 00161 Rome, Italy. [8]Department of Neuroscience, Neurosurgery Section, Università Cattolica del Sacro Cuore, Rome, Italy. [9]Institute of Human Genetics, University Medical Center Göttingen, Göttingen, Germany. [10]University of California San Diego, School of Medicine, Department of Pediatrics/Rady Children's Hospital-San Diego, San Diego, USA. [11]Department of Cellular & Molecular Medicine, Stem Cell Program, La Jolla CA 92093 MC 0695, USA. [12]Department of Biomedical Engineering, Technion-Israel Institute of Technology, Haifa, Israel. [13]Department of Clinical and Molecular Medicine, Faculty of Medicine and Health Sciences, Norwegian University of Science and Technology, Trondheim, Norway. [14]Institute of Human Genetics, University Hospital, Heinrich-Heine-Universität, 40225 Düsseldorf, Germany. [15]These authors contributed equally: Kamyar Hadian, Jay Gopalakrishnan. ✉e-mail: jay.gopalakrishnan@uni-jena.de

