## [Peer Review File · Nature Communications]

REVIEWER COMMENTS

Reviewer #1 (Remarks to the Author):

The manuscript titled "Reliability of high-quantity human brain organoids for modeling microcephaly, glioma invasion, and drug screening" by Anand Ramani and colleagues presents a method for generating high-quality human iPS-derived brain organoids (Hi-Q brain organoids). These Hi-Q brain organoids exhibit reproducible cytoarchitecture resembling diverse neural cell types, functional neural networks with synaptic activity and electrophysiological properties, and the ability to model neurodevelopmental disorder states. The Hi-Q brain organoid serves as a promising model for glioma invasion and high-throughput drug screening. Notably, these organoids demonstrate reproducibility and stability across different batches, essential criteria for high-quality brain organoids. The study is much needed and timely with a broad interest for the neuroscience community. The experimental design is well-thought and reasonable. The scientific rigor is high. On the other hand, the manuscript has some specific limitations that should be addressed.

Main Points:

1. The manuscript describes the use of AggreWell to generate spheres consistently in size and shape, along with an embedding-free method without EB formation, optimized by only using ROCK inhibitor initially. However, it remains unclear which procedure or unique aspect of the protocol contributes to producing Hi-Q brain organoids compared to previously published protocols (PMID: 33765444, PMID: 37976154, PMID: 35604169).
2. The brain organoids can recapitulate human cortex organization, it is unclear if the Hi-Q brain organoid exhibits different layers of neuron organization like in the human cortex, including ventricular zone and subventricular zone (VZ/SVZ) and outer SVZ (oSVZ).
3. The single-cell RNA-seq analysis shows that Hi-Q brain organoids have more proliferating cell types compared to EB-based organoids, which have more astrocytes. However, Figure S2 indicates that there are more neurons in EB-based organoids. Authors should reconcile these data. Does the Hi-Q brain organoid method affect neuron diversity or production?
4. Is there any difference between Hi-Q brain organoids and EB-based organoids in modeling CDK5RAP2 and CSB patient phenotypes, such as organoid size, cell death, or neural differentiation?

5. The manuscript attempts to establish a cryopreservation method, but limitations do exist. What is the recovery ratio or survival ratio after thawing Hi-Q organoids? Can this method be used for EB-based or patient-derived organoids, which may experience cell death or premature neural differentiation? The penetration of the freezing medium throughout the tissue is crucial for cryopreservation. Does the organoid size affect successful cryopreservation? The author should evaluate the cryopreservation method based on more parameters, not just cell properties in the early stages of brain organoids.

Minor Points:

1. The format of references in the literature is cited inconsistently.
2. There appears to be a mismatch between the data shown in Figures S3A and 3B and their corresponding descriptions in the text, which should be corrected to avoid confusion.

Reviewer #2 (Remarks to the Author):

Ramani et al report a method to produce brain organoid in high quantity (Hi-Q) in a relatively homogenous manner. Authors made a custom-designed spherical plate using a medical-grade, inert Cyclo-127 Olefin-Copolymer (COC). The plate contains 24 large wells, each of which has 185 microwell of 1X1 mm opening and 180 um in diameter. Authors demonstrate that the custom-made plate can produce a large number of brain organoids in homogenous manner. Then, the Hi-Q organoids were characterized their organization, cellular composition, and efficiency in modeling the genetic microcephaly and glioblastoma. While the Hi-Q method seems to produce a high quality and quantity cerebral organoids, except the custom-made plate, it is difficult to find much scientific advancement. Even the custom-made plate looks similar to the already commercially available and heavily used AggreWell? It will be essential to make a comparison of the custom-made plate with the AggreWell-based organoids, if authors want to make a comparison with the previously published works in terms of the quality of the Hi-Q organoids. Single cell data from the published works do not necessarily proper control to compare with the scRNA-seq from the Hi-Q organoids. Thus, the reduced expression of the stressed genes do not support the authors' claim that Hi-Q organoids are less stressed. Otherwise, the analysis of microcephaly and glioblastoma models were done as reported by previously published works.

Reviewer #2 (Remarks on code availability):

Codes look good and useful to the community. I was able to run the codes.

Reviewer #3 (Remarks to the Author):

In this manuscript, Ramani and colleagues reported a novel approach that can culture brain organoids with reproducible cytoarchitecture, cell diversity among different pluripotent cell lines, and tested their brain organoids in modeling genetic neural disorders as well as GBM invasion. In addition, authors also reported a cryopreservation- reculture method for brain organoid research. This manuscript aims to overcome key pitfalls in brain organoid culture, and have solved two major problems: the reproducibility between individual organoid and batches, and the re-culture after cryopreservation. Overall, the manuscript is well organized. However, there are several major concerns need to be addressed to further support their conclusion.

1. When authors test the reproducibility of Hi-Q brain organoids using scRNA-seq, they analyzed three organoids per time point of culture. They claimed that the cell diversity and proportion of different cell types are similar among different organoids. However, authors didn't examine the reproducibility regarding the cell diversity and proportion between different batches. Authors should test organoids from at least three batches for at least one time point using scRNA-seq to further support their claim.

2. Authors claim that Hi-Q method can reduce the ectopic stress-inducing pathways. Could authors give any explanation why this approach can achieve this or which treatment could potentially improve this aspect.

3. The Hi-Q method administrates neural induction medium since very beginning, while normally the EB methods started with the stem cell medium (for several days). It is difficult to understand why Hi-Q brain organoids are behind the maturation status compared to EB-derived brain organoids. Authors should give reasonable explanation.

Reliability of high-quantity human brain organoids for modeling microcephaly, glioma invasion, and drug screening

REVIEWER COMMENTS

Reviewer #1 (Remarks to the Author):

The manuscript titled "Reliability of high-quantity human brain organoids for modeling microcephaly, glioma invasion, and drug screening" by Anand Ramani and colleagues presents a method for generating high-quality human iPS-derived brain organoids (Hi-Q brain organoids). These Hi-Q brain organoids exhibit reproducible cytoarchitecture resembling diverse neural cell types, functional neural networks with synaptic activity and electrophysiological properties, and the ability to model neurodevelopmental disorder states. The Hi-Q brain organoid serves as a promising model for glioma invasion and high-throughput drug screening. Notably, these organoids demonstrate reproducibility and stability across different batches, essential criteria for high-quality brain organoids. The study is much needed and timely with a broad interest for the neuroscience community. The experimental design is well-thought and reasonable. The scientific rigor is high. On the other hand, the manuscript has some specific limitations that should be addressed.

This reviewer's comments are encouraging and help us improve the manuscript. We have addressed the comments with valid explanations, clarifications, and experiments. Our new additions in the main text are in blue.

Main Points:

1. The manuscript describes the use of AggreWell to generate spheres consistently in size and shape, along with an embedding-free method without EB formation, optimized by only using ROCK inhibitor initially. However, it remains unclear which procedure or unique aspect of the protocol contributes to producing Hi-Q brain organoids compared to previously published protocols (PMID: 33765444, PMID: 37976154, PMID: 35604169).

The mentioned works¹⁻³ have advanced the organoid generation methods. These and several other works have used commercially available AggreWell800 plates or standard Petri dishes. Here, 33765444 used AggreWell 800 plates (StemCell Technologies, containing 300 microwells, each 800 μm in size) and generated cerebral organoids via embryoid bodies with varying sizes as an intermediate¹. 37976154 used Petri dishes and generated organoids with an enriched level of oligodendrocyte differentiation². The organoids are of variable sizes. 35604169 used six-well plates, embedded the neurospheres with a basement membrane matrix, and generated cortical organoids of variable sizes. Moreover, these works have included a step where manual embedding of neurospheres may be required³.

The unique aspect of our protocol is that our method uses a custom-made plate that does not include pre-coating and centrifugation. Besides, our method does not employ embryoid body formation and Matrigel embedding before initiating the differentiation. Our method uses spinner bioreactors for culturing. In addition, our method could generate a large number of cryopreserve brain organoids with consistent size. We have cited these works in the revised version (best in the discussion section) and highlighted the critical differences between ours and the published methods.

For clarity, we provide a summary table (**Table 4**) comparing the differences between protocols that used commercially available or custom-made plates.

For further clarification, we share the following points:

The story's central question is to generate stress-free and homogeneous brain organoids suitable for various applications. The use of the microwell plate is only a part of the whole story.

We also do not exclude the possibility of making Hi-Q brain organoids in the commercially available plate. The custom-made plate may appear similar to the commercial plate (AggreWell from STEMCELL Technology) but differ remarkably in the following aspects.

-A round bottom allows cells to settle readily without any centrifugation step, which may elicit gravitational stress to iPSCs. The commercial plate has a flat bottom and requires a centrifugation step.

-A custom Plate does not require a coating step. The commercial plate requires a coating with an anti-adherence solution, which may affect the cell's physiology.

-Because one needs pre-coating and centrifugation, the number of neurosphere recoveries from the commercial plate is reduced.

2. The brain organoids can recapitulate human cortex organization, it is unclear if the Hi-Q brain organoid exhibits different layers of neuron organization like in the human cortex, including ventricular zone and subventricular zone (VZ/SVZ) and outer SVZ (oSVZ).

We appreciate this question, and the revised version has addressed it experimentally.

We previously provided SOX2 as an identity marker for VZ progenitors (**Figures 3, 5, and 6**). To address this reviewer's question, first, we analyzed our sc-RNA data. We identified the presence of additional VZ markers (such as EMX2, PAX6), SVZ markers (TBR2), and oSVZ markers (GFAP, TNC, PTPRZ1, FAM107A, HOPX, and LIFR)⁴ in variable quantities.

To experimentally validate the presence of these cell types, we stained the organoid slices for SVZ (TBR2, which has a name of EOMES) and oSVZ (PTPRZ1 and phospho-Vimentin) markers and quantified their proportions. These data are presented in new **Figure S4**, and the main text includes the interpretations. Add a conclusion sentence (e.g. Our organoids exhibit VZ, SVZ, and oSVZ, which nicely recapitulate the human cortex organization)

Fig. S4, Related to main figure 3

Figure S4: Layer identities in the ventricular zones of Hi-Q brain organoids.

A. UMAP visualization single-cell data showing six significant types of cell clusters (Color-coded).

B. Feature plots of annotated cell types positive for individual markers of VZ, SVZ, and oSVZ.

C. Dot plots displaying the marker expression levels across various age groups.

D. Immunofluorescence validation for the presence of various markers in organoid thin sections. In all cases, ARL13B (Magenta) labels primary cilia at the VZ lumen (L) at the apical side. TBR2 (Yellow) marks the distribution of intermediate progenitors, specifying the presence of SVZ, and p-Vimentin (green) marks outer radial glial cells that are basal to the VZ. P-Vimentin data in this figure should be combined with **Figure 6Ei**. Two representative illustrations from at least two independent batches of 60-day-old Hi-Q brain organoids. Scale bar 50 μm .

E. Immunofluorescence validation for the presence of PTPRZ1 (Red) specifying oSVZ. ARL13B (Magenta) labels primary cilia at the apical side of the VZ lumen (L). SOX2 (Green) marks the distribution of VZ progenitors. Two representative figures are given from at least two independent batches of organoids—scale bar 50 μm .

F. The bar diagram below shows the average number of each cell type quantified from at least six organoids from two independent batches.

3. The single-cell RNA-seq analysis shows that Hi-Q brain organoids have more proliferating cell types compared to EB-based organoids, which have more astrocytes. However, Figure S2 indicates that there are more neurons in EB-based organoids. Authors should reconcile these data. Does the Hi-Q brain organoid method affect neuron diversity or production?

This is a critical aspect to address, and we are thankful for this question. We do not intend to claim that EB generation enhances maturation. However, the differentiation steps included in those methods might positively influence maturation. Below is our reasoning, and we can adapt to other suggestions by this reviewer.

Our comparative analysis (**Now Figure S3**) shows that Hi-Q brain organoids are relatively less mature than EB-based brain organoids. This could mean that Hi-Q brain organoids contain more proliferative cell populations and that EB-based brain organoids have more differentiated cells (neurons).

Our method omits EB formation by directly exposing iPSCs to neural induction media (NIM). At this early stage, NIM does not trigger neuronal differentiation but allows the generation of pure neuroectoderm to form neural epithelia containing neural progenitors. The idea behind this strategy is to get more homogeneous neural lineages.

At the differentiation step, unlike other methods (where EB-based organoids have been generated), our method does not use retinoic acid to force neuronal maturation⁵. We also do not use any neuronal maturation factor, such as BDNF, at any point of the differentiation stage. This could explain why we observe fewer neuronal populations in Hi-Q organoids than in EB-based organoids. In other words, the Hi-Q method does not affect neuron diversity but may allow a controlled differentiation. Finally, while our method indicates that neuroectoderm formation does not require meso- and ectoderms, we can't exclude if these two germ layers impact neuronal differentiation and maturation.

In the revised version, we have highlighted these aspects and attempted to reconcile these differences better (in the discussion section).

4. Is there any difference between Hi-Q brain organoids and EB-based organoids in modeling CDK5RAP2 and CSB patient phenotypes, such as organoid size, cell death, or neural differentiation?

This reviewer has raised an important question. We have included the answers below and incorporated them into the revised version.

CDK5RAP2 model: The overall phenotype observed with CDK5RAP2 patients is similar between EB-based and Hi-Q brain organoids. The predominant phenotype is the pre-mature differentiation of progenitors into neurons. Here is the detail: Referring to Lancaster et al. 2013 (EB-based organoids) ⁶, CDK5RAP2 mutant organoids showed an altered division plane of apical progenitors. In contrast to healthy organoids, the division plane of most apical progenitors in CDK5RAP2 organoids was vertically oriented to the VZ lumen. While horizontal orientation is critical for the symmetric expansion of progenitors, vertical orientation could account for the premature differentiation of progenitors, leading to microcephaly.

We observed a similar phenotype in modeling CDK5RAP2 mutation in Hi-Q brain organoids. Notably, most apical progenitors' division planes were vertically oriented. We determined this by scoring the p-Vimentin-labeled dividing apical progenitors (**Figure 6Eii**).

CSB model: A few works modeled CSB phenotypes mainly used differentiated neurons ⁷. A recent work has characterized CSB phenotypes in brain organoids ⁸. From their description, the method of brain organoid generation is unclear. The work mainly focused on transcriptomic changes in CSB organoids, suggesting defects in brain development. Carefully examining their organoid staining data (Figure 2 in their paper), we could identify there may be some organization defects similar to what we proposed. We, however, cannot concretely conclude it since the images presented (both control and mutant) were of low magnification, insights into the cytoarchitecture were missing, and the work did not focus on DNA damage or cell death.

Our data potentially provides new insights connecting DNA damage and brain organization defects.

5. The manuscript attempts to establish a cryopreservation method, but limitations do exist. What is the recovery ratio or survival ratio after thawing Hi-Q organoids? Can this method be used for EB-based or patient-derived organoids, which may experience cell death or premature neural differentiation? The penetration of the freezing medium throughout the tissue is crucial for cryopreservation. Does the organoid size affect successful cryopreservation? The author should evaluate the cryopreservation method based on more parameters, not just cell properties in the early stages of brain organoids.

This reviewer has raised critical questions that we need to address. In the revised version, we conducted a new set of experiments, as shown below.

- We calculated the growth rate of thawed brain organoids and imaged their morphology over time (**Figure S6A-B**).
- We calculated the percentage recovery of at least eight batches of frozen Hi-Q brain organoids. Each batch contained at least ten brain organoids, and we could obtain at least 75% of the recovery after thawing. In most cases, we could recover at least 90% of the organoids (**Figure S6C**). In addition, we tested the survival by counting TUNEL-positive cells between organoids 24 hrs and ten days after thawing (**Figure S6D-E**).

- As this reviewer has pointed out, penetration of the freezing medium throughout the organoid is critical, and the size of the brain organoid probably determines it. Therefore, it is plausible that we were unsuccessful in freeze-thawing later-stage organoids (Day 35 organoids, which are larger than Day 18 organoids). Notably, these organoids did not display an intact cytoarchitectural organization after thawing. Besides, the organoids contained massively elevated TUNEL-positive cells, indicating that they did not recover after thawing (**Figure S6F**). Based on this finding, we would like to share our comments on the following reviewers' questions:

“Can this method be used for EB-based or patient-derived organoids, which may experience cell death or premature neural differentiation?”

For EB-based organoids:

While we have not tested the EB-based brain organoids, they will be challenging as they are larger than Hi-Q brain organoids. Larger in size, which will cause poor penetration of cryoprotective agents. Thus, Hi-Q brain organoids have an advantage in this aspect.

Another aspect is that Hi-Q brain organoids are relatively immature when matched with the EB-based organoids and contain more proliferating cells (**Figure S3**). This could be another advantage for a better recovery after thawing the frozen organoids.

For patient-derived organoids:

We believe that patient-derived brain organoids (generated via the Hi-Q method) are suitable for cryopreservation.

In ongoing research, we adapt the Hi-Q method and conduct high-throughput disease modeling of at least 20 neurogenetic patients. One aim of this work is to biobank patient-specific brain organoids. Our preliminary data show promising results.

We discussed these aspects in the revised manuscript.

We believe this reviewer accepts and grants our request not to conduct a whole set of new experiments with EB-based brain organoids. This would be very time consuming and may not yield incremental insight. Cryostorage also requires several months of storage before recovery.

Fig. S6 (related to main figure 5)

Figure S6: Cryopreservation, thawing, and re-culturing of Hi-Q brain organoids (Related to Figure 5)

- A.** Growth kinetics of thawed Hi-Q brain organoids (blue line) compared to control organoids (orange line) that have never been frozen. Each time point shows the average diameter of at least four organoids.
- B.** Macroscopical images of thawed Hi-Q organoids at various time points compared to control organoids that have never been frozen. Scale bar 200 μm shown in the panel.
- C.** Percentage recovery of thawed brain organoids from at least eight batches. Each batch contained at least ten brain organoids.
- D.** Cytoarchitectural analysis of thawed organoids after 48 hrs (top panel) and 10 days (bottom panel). SOX2 (magenta) specifies developing VZ, and TUNEL labels dead cells (red). The panel shows a scale bar.
- E.** Bar diagram counts TUNEL-positive cells between organoids 48 hrs and ten days after thawing of Hi-Q organoids. The number of organoids used in each experiment is denoted by “n.”
- F.** Unlike Day 18 Hi-Q organoids, thawed Day 35 organoids did not display an intact cytoarchitectural organization. In addition, thawed organoids contained massively elevated TUNEL-positive cells. TUNEL (red) labels dead cells. TUJ1 (green) labels neurons. The panel shows a scale bar.

Minor Points:

1. The format of references in the literature is cited inconsistently.

We have corrected this oversight and cited the references correctly.

2. There appears to be a mismatch between the data shown in Figures S3A and 3B and their corresponding descriptions in the text, which should be corrected to avoid confusion.

Figure 3B shows cortical markers specified by DCX, MAP2, TUJ1, Tau, and PCP4 in Day 60 brain organoids. We extended this data for Day 20 (**Figure S4A**) and Day 60 (**Figure S4B**) organoids, displaying the presence of PSD95, Synapsin 1, and CTIP2. We hope we have clarified this issue.

Reviewer #2 (Remarks to the Author):

Ramani et al report a method to produce brain organoid in high quantity (Hi-Q) in a relatively homogenous manner. Authors made a custom-designed spherical plate using a medical-grade, inert Cyclo-127 Olefin-Copolymer (COC). The plate contains 24 large wells, each of which has 185 microwell of 1X1 mm opening and 180 um in diameter. Authors demonstrate that the custom-made plate can produce a large number of brain organoids in homogenous manner. Then, the Hi-Q organoids were characterized their organization, cellular composition, and efficiency in modeling the genetic microcephaly and glioblastoma. While the Hi-Q method seems to produce a high quality and quantity cerebral organoids, except the custom-made plate, it is difficult to find much scientific advancement. Even the custom-made plate looks similar to the already commercially available and heavily used AggreWell? It will be essential to make a comparison of the custom-made plate with the AggreWell-based organoids, if authors want to make a comparison with the previously published works in terms of the quality of the Hi-Q organoids.

We are afraid there was a misunderstanding. We don't want to claim that the organoids made using our custom-made plate are superior to those made with commercial plates. The story's central question is to generate stress-free and homogeneous brain organoids suitable for various applications. The use of the microwell plate is only a part of the whole story.

We also do not exclude the possibility of making Hi-Q brain organoids in the commercially available plate. The custom-made plate may appear similar to the commercial plate (AggreWell from STEMCELL Technology) but differ remarkably in the following aspects.

-A round bottom allows cells to settle readily without any centrifugation step, which may elicit gravitational stress to iPSCs. The commercial plate has a flat bottom and requires a centrifugation step.

-A custom Plate does not require a coating step. The commercial plate requires a coating with an anti-aggregate solution, which may affect the cell's physiology.

-Because one needs pre-coating and centrifugation, the number of neurosphere recoveries from the commercial plate is reduced.

Indeed, we have compared both plates side by side. We noticed that iPSCs readily settled within a day of plating in our spherical plate, even without a centrifugation step. In contrast, the iPSCs did not settle well in commercially available microwell plates that required pre-coating. This finding suggests that our spherical plate may provide a more suitable environment for sphere formation (**Figure 1, attached**).

Second is the recovery rate of neurospheres. We could transfer the neurospheres from the custom-made plate to the spinner flask. We could recover more than 90% of neurospheres from the round-bottom (custom-made) plate. We lost at least 20% of neurospheres, mostly placed at the peripheral microwells of the commercial plate (**Figure 1, attached**).

We did not include these direct comparisons as they sound too aggressive. Besides, we don't want to claim that the organoids made via our custom-made plate are superior to those made with commercial plates. We also do not exclude the possibility of making Hi-Q brain organoids in the commercially available plate. Thus, doing a whole set of new experiments using these two plates is too time-consuming and out of the scope.

Nevertheless, for clarity, we have provided a summary table comparing various methods using various microwell plates (**Table 4**).

Single cell data from the published works do not necessarily proper control to compare with the scRNA-seq from the Hi-Q organoids. Thus, the reduced expression of the stressed genes do not support the authors' claim that Hi-Q organoids are less stressed.

To match and assess where our Hi-Q organoids fit, it was essential to use an unbiased method to compare and contrast. We chose to compare scRNA seq data. It is a usual practice, as several works have used this approach. We would adapt if this reviewer proposed an alternative method to compare.

To support our computational data, we have generated experimental data for the direct observation of cell stress by staining specific ER-stress markers (GORASP2 used by Bhaduri et al., Nature 2020)⁹ (**Figure 2, attached**).

As one can see, Hi-Q brain organoids do not show GORASP2 expression, which is in striking contrast with organoids shown in Bhaduri et al., Nature 2020⁹. Besides, the organoids shown in Bhaduri et al. have damaged cytoarchitecture.

However, Hi-Q brain organoids show a massive elevation of GORASP2-positive cells only after hydrogen peroxide treatment, which we used as a control to induce stress. We would prefer not to include this data as the Hi-Q brain organoids do not show GORASP2-positive cells, and it remains complex what effects hydrogen peroxide causes. However, if the reviewer insists, we could also add this data.

Figure 1

Custom-designed spherical plate, Round bottom

Commercial plate, Flat bottom

Figure 2

Figure legend:

Figure 1: Comparison of iPSCs behavior between custom-made spherical and flat-bottom commercial plates. iPSCs settle and form a spherical shape in the custom-made plate without centrifugation or pre-coating. Yellow arrows point to the dispersed iPSCs that do not settle well in a commercial plate. The bar diagram below quantifies the neurosphere formation and recovery between custom and commercial plates.

Figure 2: Under normal conditions, Hi-Q brain organoids do not show GORASP2-positive (red) cells. SOX2 (green) shows the cytoarchitecture of a typical ventricular zone.

Hi-Q brain organoids show elevated levels of GORASP2-positive (red) cells only when cellular stress is induced using hydrogen peroxide.

Otherwise, the analysis of microcephaly and glioblastoma models were done as reported by previously published works.

Modeling microcephaly (two different kinds) and glioma invasion assays (**Figures 6 and 7**) were to prove the broader applicability and validation of Hi-Q brain organoids as a functional 3D system. Besides glioma invasion assays, we have also provided proof-of-principle experiments to show that Hi-Q brain organoids can be used for drug screening approaches. It is the most wanted application in the field of brain organoids and personalized medicine. We now show that employing Hi-Q brain organoids for high throughput modeling of rare genetic diseases is also possible. We have discussed all of these in our discussion section.

Reviewer #2 (Remarks on code availability):

Codes look good and useful to the community. I was able to run the codes.

We are happy about it.

Reviewer #3 (Remarks to the Author):

In this manuscript, Ramani and colleagues reported a novel approach that can culture brain organoids with reproducible cytoarchitecture, cell diversity among different pluripotent cell lines, and tested their brain organoids in modeling genetic neural disorders as well as GBM invasion. In addition, authors also reported a cryopreservation- reculture method for brain organoid research. This manuscript aims to overcome key pitfalls in brain organoid culture, and have solved two major problems: the reproducibility between individual organoid and batches, and the re-culture after cryopreservation. Overall, the manuscript is well organized. However, there are several major concerns need to be addressed to further support their conclusion.

We are encouraged to hear this reviewer's comments, and addressing them improves the manuscript. Our new additions in the main text are in blue.

1. When authors test the reproducibility of Hi-Q brain organoids using scRNA-seq, they analyzed three organoids per time point of culture. They claimed that the cell diversity and proportion of different cell types are similar among different organoids. However, authors didn't examine the reproducibility regarding the cell diversity and proportion between different batches. Authors should test organoids from at least three batches for at least one time point using scRNA-seq to further support their claim.

To address this critical comment, we have conducted a new experiment in which we generated and sequenced Day 25 Hi-Q brain organoids across three independent batches and compared them. To test the similarities in cell diversity, we examined scRNA data sets of organoids cultured from three independent batches. These analyses now formed a new figure (**Figure S1**).

We processed the raw data for mapping, quantitation, and downstream analysis and normalized it before log transformation (Described in the method section). To assess the similarities between the batches, we standardized the comparison using the 2000 most highly variable genes and calculated on the uncorrected data. Our *k*-nearest neighbor network (*knn*) approach used a PCA embedding (**Figure S1 A-B**). In this analysis, the cells did not cluster by batches, suggesting the degree of similarity is high or the presence of low covariance in batch variation, and hence no correction was required for batch-to-batch variation.

To ease the analysis of the similarities between batches, we chose three major cell types: progenitors (based on SOX2, GLI3, and PAX6), cycling progenitors (based on MKI67, CENPF, and NUSAP1), and early neurons (based on DCX, NCAM, and GAP43). We then analyzed the differences between the proportions of cells in each type (**Figure S1 C-E**). This analysis did not yield significant differences, indicating a low batch-to-batch variation in terms of cell types and proportions in our Hi-Q brain organoids.

Fig. S1

Figure S1: Testing batch-to-batch variations in Hi-Q brain organoids: **A-B.** Diagnostic **(A)** and a violin plot **(B)** of principal component (PC) analysis of sc-RNA transcriptomes from three independent batches of Hi-Q brain organoids representing cells. The batch effect is not apparent as all medians of samples are close to zero. **C.** UMAP visualization and annotated cell types in three different batches. **D.** Proportions of cells in each type showing no significant differences, confirming low batch-to-batch variation in terms of cell types and their proportions. **E.** Feature plot of cells positive for individual marker genes used to identify and annotate cell types.

2. Authors claim that Hi-Q method can reduce the ectopic stress-inducing pathways. Could authors give any explanation why this approach can achieve this or which treatment could potentially improve this aspect.

The observed effect could be due to several factors combined. First, the Hi-Q method minimizes manual handling of iPS cells, such as embedding them with Matrigel and incubating them in various dishes before transferring them to spinner flasks. Second, the Hi-Q method does not use the ROCK inhibitor throughout the culturing. We use it only on day 1 of the iPS seeding.

While using a Rho-kinase (ROCK) inhibitor at a concentration of 10 μ M at an early stage of iPSCs seeding can alleviate cell death, prolonged exposure and/or higher concentrations could change the cell's metabolism and induce the meso-endodermal differentiation pathway^{10,11}. Indeed, prolonged use of ROCK inhibitors is associated with generating organoids with ectopically active cellular stress pathways⁹. Therefore, after 24 hours of initial culturing in a neural induction medium, we omitted the ROCK inhibitor. Third, with the custom-made plate, we do not require precoating the plates or centrifuging the cells, which will avoid stress due to gravitational force. We have mentioned these factors in the manuscript (best in the discussion section). Fourth, the rotary suspension culture in high medium volume could reduce sheer stress than the shaker methods in low medium volume. Here, organoids roll on the vessels' surface not suspending in the medium. We have them highlighted in the revised version.

3. The Hi-Q method administrates neural induction medium since very beginning, while normally the EB methods started with the stem cell medium (for several days). It is difficult to understand why Hi-Q brain organoids are behind the maturation status compared to EB-derived brain organoids. Authors should give reasonable explanation.

We are thankful for this question, and it is intriguing.

We do not intend to claim that EB generation enhances maturation. However, the differentiation steps included in those methods might influence the maturation positively. Below is our reasoning, and we can adapt to other suggestions by this reviewer.

Our comparative analysis **(Figure S3)** shows that Hi-Q brain organoids are relatively less mature than some published EB-based brain organoids. This could mean that Hi-Q brain organoids contain more proliferative cell populations and that EB-based brain organoids have more differentiated cells (neurons).

Our method omits EB formation by directly exposing iPSCs to neural induction media (NIM). At this early stage, NIM does not trigger neuronal differentiation but allows the generation of pure neuroectoderm to form neural epithelia containing neural progenitors. The idea behind this strategy is to achieve more homogeneous neural lineages.

At the differentiation step, unlike other methods (where EB-based organoids have been generated), our method does not use retinoic acid to force neuronal maturation⁵. We also do not use any neuronal maturation factor, such as BDNF or GDNF, at any point of the differentiation stage. This could explain why we observe fewer neuronal populations in Hi-Q organoids than in EB-based organoids. In other words, the Hi-Q organoids are behind the maturation status compared to EB-derived brain organoids. However, the Hi-Q organoids' generation method does not affect neuron diversity but may allow a controlled differentiation. Finally, while our method indicates that neuroectoderm formation does not require meso- and ectoderms, we can't exclude if these two germ layers impact neuronal differentiation.

In the revised version, we have explained these potential reasons.

- 1 Benito-Kwiecinski, S. *et al.* An early cell shape transition drives evolutionary expansion of the human forebrain. *Cell* **184**, 2084-2102.e2019 (2021). <https://doi.org:10.1016/j.cell.2021.02.050>
- 2 Al-Mhanawi, B. *et al.* Protocol for generating embedding-free brain organoids enriched with oligodendrocytes. *STAR Protoc* **4**, 102725 (2023). <https://doi.org:10.1016/j.xpro.2023.102725>
- 3 Shaker, M. R., Hunter, Z. L. & Wolvetang, E. J. Robust and Highly Reproducible Generation of Cortical Brain Organoids for Modelling Brain Neuronal Senescence In Vitro. *Journal of visualized experiments : JoVE* (2022). <https://doi.org:10.3791/63714>
- 4 Pollen, A. A. *et al.* Molecular identity of human outer radial glia during cortical development. *Cell* **163**, 55-67 (2015). <https://doi.org:10.1016/j.cell.2015.09.004>
- 5 Siegenthaler, J. A. *et al.* Retinoic acid from the meninges regulates cortical neuron generation. *Cell* **139**, 597-609 (2009). <https://doi.org:10.1016/j.cell.2009.10.004>
- 6 Lancaster, M. A. *et al.* Cerebral organoids model human brain development and microcephaly. *Nature* **501**, 373-379 (2013). <https://doi.org:10.1038/nature12517>
- 7 Vessoni, A. T. *et al.* Cockayne syndrome-derived neurons display reduced synapse density and altered neural network synchrony. *Human molecular genetics* **25**, 1271-1280 (2016). <https://doi.org:10.1093/hmg/ddw008>
- 8 Szepanowski, L. P. *et al.* Cockayne Syndrome Patient iPSC-Derived Brain Organoids and Neurospheres Show Early Transcriptional Dysregulation of Biological Processes Associated with Brain Development and Metabolism. *Cells* **13** (2024). <https://doi.org:10.3390/cells13070591>
- 9 Bhaduri, A. *et al.* Cell stress in cortical organoids impairs molecular subtype specification. *Nature* **578**, 142-148 (2020). <https://doi.org:10.1038/s41586-020-1962-0>
- 10 Maldonado, M., Luu, R. J., Ramos, M. E. & Nam, J. ROCK inhibitor primes human induced pluripotent stem cells to selectively differentiate towards mesendodermal lineage via epithelial-mesenchymal transition-like modulation. *Stem cell research* **17**, 222-227 (2016). <https://doi.org:10.1016/j.scr.2016.07.009>
- 11 Le, M. N. T. & Hasegawa, K. Expansion Culture of Human Pluripotent Stem Cells and Production of Cardiomyocytes. *Bioengineering (Basel)* **6** (2019). <https://doi.org:10.3390/bioengineering6020048>

REVIEWERS' COMMENTS

Reviewer #1 (Remarks to the Author):

The authors have fully addressed my comments through additional experiments and provided clear results. The revisions have strengthened the manuscript, and all concerns raised during the previous review have been resolved. Based on these improvements, I recommend that the manuscript be accepted for publication.

Reviewer #2 (Remarks to the Author):

Although the revised manuscript addresses the comments, still except the custom-made plate, which seems very similar to ultra low attachment plat from Fisher, findings in the manuscript seem similar to already published other works combined.

Reviewer #3 (Remarks to the Author):

Authors have addressed all my concerns in the revised manuscript.